# Open Materials Generation with Inference-Time Reinforcement Learning

**Philipp Höllmer** [1]    **Stefano Martiniani** [1]

## Abstract

Continuous-time generative models for crystalline materials enable inverse materials design by learning to predict stable crystal structures, but incorporating explicit target properties into the generative process remains challenging. Policy-gradient reinforcement learning (RL) provides a principled mechanism for aligning generative models with downstream objectives but typically requires access to the score, which has prevented its application to flow-based models that learn only velocity fields. We introduce Open Materials Generation with Inference-time Reinforcement Learning (OMatG-IRL), a policy-gradient RL framework that operates directly on the learned velocity fields and eliminates the need for the explicit computation of the score. OMatG-IRL leverages stochastic perturbations of the underlying generation dynamics preserving the baseline performance of the pretrained generative model while enabling exploration and policy-gradient estimation at inference time. Using OMatG-IRL, we present the first application of RL to crystal structure prediction (CSP). Our method enables effective reinforcement of an energy-based objective while preserving diversity through composition conditioning, and it achieves performance competitive with score-based RL approaches. Finally, we show that OMatG-IRL can learn time-dependent velocity-annealing schedules, enabling accurate CSP with order-of-magnitude improvements in sampling efficiency and, correspondingly, reduction in generation time. The OMatG-IRL code is included in a new release of the Open Materials Generation (OMatG) framework available at https://github.com/FERMat-ML/OMatG.

[1]New York University. Correspondence to: Stefano Martiniani <sm7683@nyu.edu>.

*Proceedings of the 43rd International Conference on Machine Learning*, Seoul, South Korea. PMLR 306, 2026. Copyright 2026 by the author(s).

## 1. Introduction

The discovery and development of novel crystalline materials with targeted properties are fundamental to technological progress and machine-learning methods are rapidly emerging as powerful tools of modern materials discovery (Sanchez-Lengeling & Aspuru-Guzik, 2018; Schmidt et al., 2019; Noh et al., 2020; Liu et al., 2023; Merchant et al., 2023; Park et al., 2024; Handoko & Made, 2025; De Breuck et al., 2025; Ding et al., 2025; Zhang et al., 2026; Abhyankar et al., 2026; Chaudhari et al., 2026). Central to this development are generative models capable of realizing an inverse-design approach by suggesting stable and novel crystal structures with predefined target properties.

Generative models for crystalline materials learn the chemical rules underlying crystal-structure datasets by approximating the high-dimensional distribution from which the data is drawn. Since the datasets typically comprise experimentally realized or computationally relaxed structures (Xie et al., 2022; Zeni et al., 2025), chemical (meta-)stability is implicitly learned as the primary target property of the generated samples. Accordingly, generative models are commonly benchmarked with respect to (meta-)stability. In the *de novo generation* (DNG) task, where crystal structure and composition are generated jointly, stability is typically evaluated directly (Betala et al., 2025; Szymanski & Bartel, 2025). In the *crystal structure prediction* (CSP) task, where a structure is generated for a given composition, stability is instead assessed by proxy: Computationally efficient structure-matching metrics such as match rate (Xie et al., 2022) and METRe/cRMSE (Martirossyan et al., 2025) compare generated structures to known stable structures.

Beyond the implicit stability requirement, generative models in an inverse materials-design pipeline should also be able to align generation with explicit target properties such as desired mechanical, electronic, and magnetic properties. *Reinforcement learning* (RL) provides a flexible framework for directly optimizing generative models with respect to downstream objectives using black-box reward functions (Fan & Lee, 2023; Black et al., 2024). These rewards can also render implicit constraints such as chemical (meta-)stability explicit for systematic optimization. More broadly, RL overcomes a fundamental limitation of likelihood-based training of generative models, as it prioritizes task-specific objec-

tives rather than the accurate approximation of the data log-likelihood. RL has proven highly effective, for example, for large language models, computer vision, and molecule design (Cao et al., 2025). In contrast, the application of RL to generative models for crystalline materials remains comparatively underexplored (De Breuck et al., 2025).

## 1.1. Related Works

A rapidly growing body of work has developed generative frameworks for inorganic crystalline materials using a wide range of representations, architectures, and generative paradigms [see, e.g., the recent review by De Breuck et al. (2025)]. An important class of continuous-time approaches considers joint generative processes over atomic (fractional) coordinates, lattice parameters, and atom types to produce periodic structures in real space without explicit space-group constraints [for symmetry-aware variants we refer to De Breuck et al. (2025)]. Many of these frameworks rely on score-based diffusion models (Sohl-Dickstein et al., 2015; Ho et al., 2020; Song et al., 2021) which learn the gradient of the log probability density and generate samples via reverse diffusion processes (Jiao et al., 2023; Yang et al., 2024; Zeni et al., 2025; Joshi et al., 2025; Cornet et al., 2025; Tangsongcharoen et al., 2025; Park et al., 2025; Das et al., 2025; Khastagir et al., 2026). In contrast, flow-matching models (Lipman et al., 2023; Albergo & Vanden-Eijnden, 2023; Chen & Lipman, 2024; Lipman et al., 2024) learn a velocity field and generate samples by integrating an ordinary differential equation (Miller et al., 2024; Sriram et al., 2024; Luo et al., 2025). [Bayesian flow networks (Graves et al., 2025) have also recently been adapted to periodic crystal generation (Wu et al., 2025; Ruple et al., 2026).] Finally, OMatG (Höllmer et al., 2025) builds on *stochastic interpolants* (SI) (Albergo et al., 2025), a unifying framework that encompasses both flow-matching and diffusion-based methods as special cases through specific choices of interpolants. Flow-matching and SI models can initiate generation from arbitrary base distributions, which has been shown to substantially improve performance (Miller et al., 2024; Sriram et al., 2024; Höllmer et al., 2025). Moreover, they typically require significantly fewer integration steps during inference than diffusion-based sampling, leading to markedly higher sampling efficiency (Miller et al., 2024; Luo et al., 2025; Höllmer et al., 2025).

Some of these frameworks are able to align generation with explicit target properties. This is typically achieved via direct input augmentation to the underlying network, either in a single conditional model (Luo et al., 2025), or by combining unconditional and conditional models through classifier-free guidance (Ho & Salimans, 2022; Yang et al., 2024; Zeni et al., 2025; Tangsongcharoen et al., 2025; Prakash et al., 2026). These approaches require sufficiently large and diverse labeled datasets for the desired target properties

and remain constrained by the support of the underlying training distribution. Only recently, MatInvent (Chen et al., 2025) and Chemeleon2 (Park & Walsh, 2026) applied policy-gradient RL frameworks to the DNG task, with the latter adopting group-relative policy optimization (GRPO) (Shao et al., 2024) in conjunction with proximal policy optimization (PPO) (Schulman et al., 2017) to ensure stable optimization. Notably, both approaches had to incorporate explicit diversity rewards to prevent mode collapse during training.

Both MatInvent and Chemeleon2 apply RL to (latent) diffusion models, which provide explicit access to the score required by policy-gradient RL methods. Flow-matching frameworks and much of the SI framework in OMatG, however, only learn velocity fields without computing the score. Flow-GRPO addresses this limitation for the specific case of Gaussian base distributions and linear interpolant paths, where the score can be related to the learned velocity field (Liu et al., 2025). Still, a substantial portion of the SI design space in OMatG, whose systematic exploration previously led to state-of-the-art benchmark results, remains inaccessible to RL. This limitation motivates the development of RL methods that operate across the full SI design space without requiring explicit score representations.

## 1.2. Our Contribution

We introduce *Open Materials Generation with Inference-time Reinforcement Learning* (OMatG-IRL), a policy-gradient RL framework for continuous-time flow-based generative models, and apply it to the CSP task. Our method operates directly on the learned velocity field and does not require an explicit score computation, while also naturally extending to score-based models when available. OMatG-IRL is inspired by the empirical observation that standard CSP evaluation metrics are robust to the introduction of stochastic perturbations into the underlying ordinary differential equation, enabling their use for exploration and policy-gradient estimation (see Fig. 1).

Taking advantage of OMatG's flexibility to learn both velocity fields and scores, we directly compare score-based and velocity-based OMatG-IRL and demonstrate that both achieve comparable reinforcement performance.

This work also presents the first application of RL to the CSP task. We show that energy-based objectives can be effectively reinforced without the need for explicit diversity rewards; in contrast to DNG models, diversity in the CSP task naturally emerges from conditioning on composition.

Finally, we demonstrate that our RL framework can be used to learn a time-dependent velocity-annealing schedule, replacing handcrafted annealing schemes and enabling accurate CSP with an order-of-magnitude reduction in the number of integration steps.

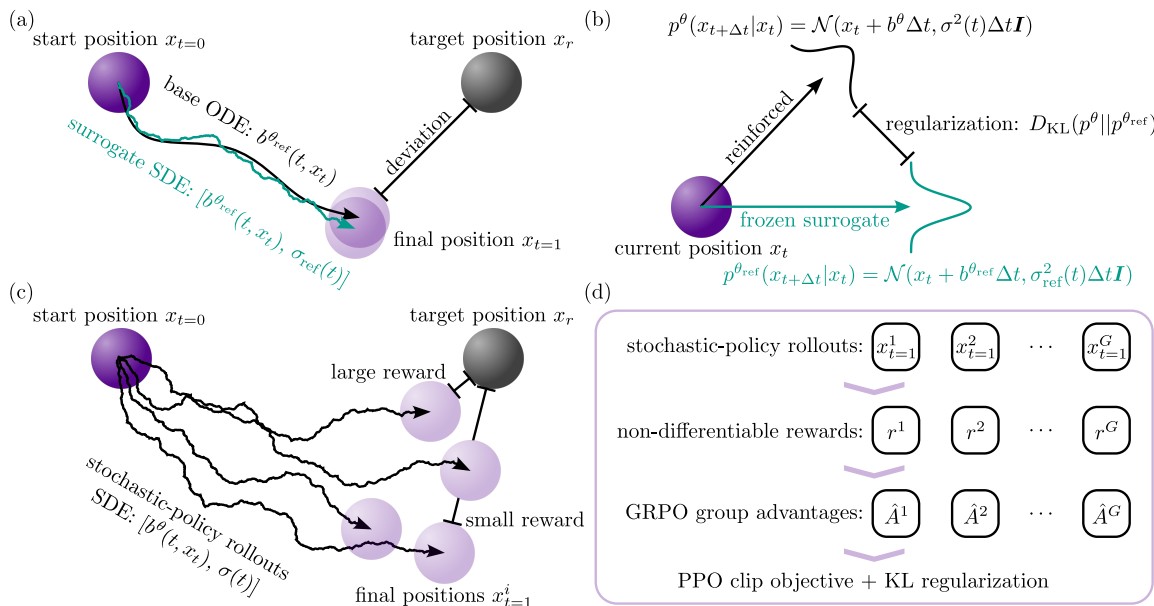

*Figure 1.* Inference-time RL for CSP in velocity-based OMatG-IRL. (a) The deterministic base ODE with pretrained velocity field $b^{\theta_{\text{ref}}}(t, x_t)$ is augmented with a small noise schedule $\sigma_{\text{ref}}(t)$, yielding a surrogate SDE that leaves evaluation metrics (e.g., deviation from a reference structure) of the final samples $x_{t=1}$ virtually unchanged. (b) The frozen surrogate defines a reference policy for KL regularization, while stochastic exploration is performed using a reinforced velocity field $b^\theta(t, x_t)$ and a (potentially different) noise schedule $\sigma(t)$. (c) GRPO compares terminal rewards $r^i = r(x_{t=1}^i)$ obtained from multiple stochastic-policy rollouts under identical conditioning. (d) These rewards are transformed into GRPO group advantages and used, together with KL regularization, in a PPO-style clipped objective to update the policy.

## 2. Background

We implement OMatG-IRL within the open-source OMatG framework. Here, we briefly review the key components of OMatG that are relevant for the CSP task, i.e., its crystal representation and SI-based generative process (see Sections 2.1 and 2.2). We then introduce the formulation of policy-gradient RL for continuous-time stochastic generative models (see Section 2.3).

### 2.1. Equivariant Crystal Representation

OMatG represents a crystalline material by its unit cell, which serves as the fundamental building block of the infinite periodic crystal. A unit cell containing $N$ atoms is represented by three components $\{\boldsymbol{A}, \boldsymbol{X}, \boldsymbol{L}\}$. Here, $\boldsymbol{A} \in \mathbb{N}_{>0}^N$ denotes the atomic numbers of the atoms, $\boldsymbol{X} \in [0, 1)^{N \times 3}$ their fractional coordinates on a torus, and $\boldsymbol{L} \in \mathbb{R}^{3 \times 3}$ the lattice matrix whose rows correspond to the three lattice vectors. The lattice vectors span a parallelepiped with volume $|\det \boldsymbol{L}| > 0$. Cartesian coordinates of the atoms within the parallelepiped are obtained as $\boldsymbol{X}\boldsymbol{L}$, and periodic replication of the unit cell generates the infinite crystal.

The crystal structure is invariant under joint permutations of atom indices in $\boldsymbol{A}$ and $\boldsymbol{X}$, global rotations of the lattice (which induce rigid rotations of the entire crystal), and translations of the fractional coordinates $\boldsymbol{X}$ (modulo 1). OMatG

employs CSPNet (Jiao et al., 2023) as its model architecture, an $E(n)$-equivariant graph neural network (Satorras et al., 2021) whose internal representation is equivariant to permutations and rotations, and invariant to translations. As a result, the outputs of the neural network preserve the same symmetries.

### 2.2. Stochastic Interpolants

OMatG adopts the SI framework to model the continuous lattice matrices $\boldsymbol{L}$ and fractional coordinates $\boldsymbol{X}$. For the CSP task, the atomic numbers $\boldsymbol{A}$ are solely provided as fixed conditional inputs during the generative process. While the model output depends on the full triplet $\{\boldsymbol{A}, \boldsymbol{X}, \boldsymbol{L}\}$ during both training and generation, only the continuous variables $\boldsymbol{X}$ and $\boldsymbol{L}$ are interpolated and integrated in parallel by the SI dynamics. In the following, we use the symbol $x$ to denote a generic continuous configuration variable, that is, either $\boldsymbol{X}$ or $\boldsymbol{L}$, with the understanding that model inputs always refer to the full triplet.

A stochastic interpolant defines a stochastic process that bridges an arbitrary base distribution $\rho_0$ and the data distribution $\rho_1$ (Albergo et al., 2025):

$$x_t := x(t, x_0, x_1, z) = \alpha(t)\, x_0 + \beta(t)\, x_1 + \gamma(t)\, z. \quad (1)$$

Here, the time $t \in [0, 1]$ evolves a continuous random variable $x_t$ from a sample $x_0 \sim \rho_0$ from the base distribution to

a sample $x_1 \sim \rho_1$ from the training data. This construction requires, among other constraints (Albergo et al., 2025), $\alpha(0) = \beta(1) = 1$ and $\alpha(1) = \beta(0) = \gamma(0) = \gamma(1) = 0$. The random variable $z$ is drawn from a standard Gaussian $\mathcal{N}(0, \boldsymbol{I})$ and injects stochasticity into the interpolant path.

The conditional velocity $\partial_t x_t$ from Eq. (1) is used to train an unconditional velocity field $b^\theta(t, x_t)$ that depends only on the current time $t$ and configuration $x_t$ by minimizing the loss function

$$\mathcal{L}_b(\theta) = \mathbb{E}_{t,z,x_0,x_1} \left\| b^\theta(t, x_t) - \partial_t x_t \right\|^2. \qquad (2)$$

The expectation is taken independently over $z \sim \mathcal{N}(0, \boldsymbol{I})$, $x_0 \sim \rho_0$, $x_1 \sim \rho_1$, and $t \sim \mathcal{U}(0, 1)$ where $\mathcal{U}(0, 1)$ is the uniform distribution between $0$ and $1$. Remarkably, the minimizer of Eq. (2) yields a velocity field whose marginal distributions $\rho_t$ match those of the stochastic process $x_t$ in Eq. (1). In particular, integrating the ordinary differential equation (ODE)

$$\mathrm{d}X_t = b^\theta(t, X_t) \, \mathrm{d}t \qquad (3)$$

from an initial sample $x_0 \sim \rho_0$ at $t = 0$ to $t = 1$ produces a sample $x_1$ from the data distribution $\rho_1$. This ODE is usually numerically integrated with the forward Euler method with $N_t$ time steps of size $\Delta t$:

$$x_{t+\Delta t} = x_t + b^\theta(t, x_t) \, \Delta t. \qquad (4)$$

In addition to the velocity field, one can optionally learn to predict the denoiser $z^\theta(t, x_t)$ at a given time $t$ and configuration $x_t$ by minimizing the loss function

$$\mathcal{L}_z(\theta) = \mathbb{E}_{t,z,x_0,x_1} \left\| z^\theta(t, x_t) - z \right\|^2. \qquad (5)$$

The denoiser $z^\theta(t, x_t)$ is related to the score via $\nabla \log \rho^\theta(t, x_t) = -z^\theta(t, x_t)/\gamma(t)$. Equation (5) enables generative modeling by integrating the stochastic differential equation (SDE)

$$\mathrm{d}X_t = \left[ b^\theta(t, X_t) - \frac{\sigma^2(t)}{2\gamma(t)} z^\theta(t, X_t) \right] \mathrm{d}t + \sigma(t) \, \mathrm{d}W_t, \quad (6)$$

where $\mathrm{d}W_t$ denotes Wiener process increments and $\sigma(t)$ controls the stochasticity. This SDE is usually numerically integrated from $t = 0$ to $t = 1$ with $N_t$ Euler–Maruyama updates of size $\Delta t$:

$$\begin{aligned} x_{t+\Delta t} = x_t + \left[ b^\theta(t, x_t) - \frac{\sigma^2(t)}{2\gamma(t)} z^\theta(t, x_t) \right] \Delta t \\ + \sigma(t)\sqrt{\Delta t}\, \xi, \end{aligned} \qquad (7)$$

where $\xi \sim \mathcal{N}(0, \boldsymbol{I})$. This update induces an isotropic Gaussian conditional distribution $p^\theta(x_{t+\Delta t}|x_t)$ with the mean given by the drift term and covariance $\sigma^2(t)\Delta t\, \boldsymbol{I}$.

Specific choices of the stochastic interpolants in Eq. (1), combined with appropriate ODE- or SDE-based sampling schemes, recover flow-matching and score-based diffusion models as special cases. The generality of the SI framework, however, enables exploration of a substantially larger design space.

## 2.3. Policy-Gradient Reinforcement Learning

To apply policy-gradient RL to *stochastic* generative models, we formulate their iterative numerical integration as a Markov decision process $(\mathcal{S}, \mathcal{A}, \phi_0, P, R)$ with state space $\mathcal{S}$, action space $\mathcal{A}$, initial-state distribution $\phi_0$, transition kernel $P$, and reward function $R$ (Black et al., 2024). The state at time $t$ is $s_t := (t, x_t) \in \mathcal{S}$, and the initial-state distribution is $\phi_0 := (\delta_0, \rho_0)$, where $\delta_y$ denotes a Dirac delta distribution centered at $y$. At each time step, an agent samples an action $a_t \in \mathcal{A}$ from its policy $\pi^\theta(a_t|s_t) := p^\theta(x_{t+\Delta t}|x_t)$, corresponding to selecting the next configuration $a_t := x_{t+\Delta t}$. The environment transition is deterministic: $P(s_{t+\Delta t}|s_t, a_t) := (\delta_{t+\Delta t}, \delta_{x_{t+\Delta t}})$. Rewards are assigned only at the terminal state, with $R(s_t, a_t) := r(s_{t=1})$ if $t = 1$ and $R(s_t, a_t) := 0$ otherwise, where $r(s) = r(x)$ is an arbitrary configuration-dependent black-box reward function. The policy-gradient RL objective is to maximize the expected terminal reward over trajectories $\tau = (s_0, a_0, \ldots, s_1, a_1)$ sampled from the policy (denoted here as $\tau \sim \pi^\theta$):

$$\mathcal{J}_{\mathrm{RL}}(\theta) = \mathbb{E}_{\tau \sim \pi^\theta} \, r(s_{t=1}). \qquad (8)$$

To update the current policy $\pi^{\theta_{\mathrm{old}}} \to \pi^\theta$ to maximize $\mathcal{J}_{\mathrm{RL}}(\pi^\theta)$, we adopt GRPO (Shao et al., 2024), a policy-gradient method that directly compares rewards across multiple trajectories produced under identical conditioning. For the CSP task, this corresponds to rolling out $G$ trajectories $\{\tau^i \sim \pi^{\theta_{\mathrm{old}}}\}_{i=1}^G$ for the same composition, which may or may not start from different initial configurations $x_{t=0}^i$. The terminal rewards $r^i = r(x_{t=1}^i)$ are then compared within the group to compute group-relative advantages

$$\hat{A}^i = \frac{r^i - \mathrm{mean}(\{r^i\}_{i=1}^G)}{\mathrm{std}(\{r^i\}_{i=1}^G)}. \qquad (9)$$

Since rewards are only defined at the terminal time, the same group-relative advantage $\hat{A}^i$ is used for all time steps along trajectory $\tau^i$. The RL objective in Eq. (8) is then optimized by maximizing a surrogate objective inspired by PPO (Schulman et al., 2017), averaged over the group and all time steps:

$$\begin{aligned} \mathcal{J}_{\mathrm{GRPO}}(\theta) = \frac{\alpha}{GN_t} \sum_{i,t} \min \Big[ q_t^i(\theta)\hat{A}^i, \\ \mathrm{clip}\big(q_t^i(\theta), \varepsilon\big)\hat{A}^i \Big]. \end{aligned} \qquad (10)$$

Here, $\alpha$ is a weighting coefficient, $\text{clip}(x, \varepsilon) := \text{clip}(x, 1 - \varepsilon, 1 + \varepsilon)$ clamps $x$ to the interval $[1 - \varepsilon, 1 + \varepsilon]$, and $q_t^i(\theta) := p^\theta(x_{t+\Delta t}^i | x_t^i) / p^{\theta_{\text{old}}}(x_{t+\Delta t}^i | x_t^i)$ is the policy ratio between the updated policy and the old policy that generated the trajectories. Since crystal structures typically contain variable numbers of atoms, we consider different normalization strategies for the structure-level terms in the surrogate-objective sum in Eq. (10) to prevent policy updates from being biased toward larger crystals, and treat the choice of strategy as a hyperparameter (see Appendix A). Given a fixed group of trajectories, the objective in Eq. (10) is typically optimized for multiple gradient steps (referred to as PPO epochs). To stabilize training, the clipping operation limits large policy updates in a trust-region-like manner controlled by the hyperparameter $\varepsilon$. The group-relative normalization of the advantages in Eq. (9) removes the need for a learned value function and stabilizes optimization across heterogeneous reward scales.

Typically, one includes an additional KL-regularization term in the maximized objective:

$$\mathcal{J}_{\text{KL}}(\theta) = -\frac{\beta}{GN_t} \sum_{i,t} D_{\text{KL}} \big[ p^\theta(x_{t+\Delta t}^i | x_t^i) \, || \tag{11}$$
$$p^{\theta_{\text{ref}}}(x_{t+\Delta t}^i | x_t^i) \big].$$

Here, $D_{\text{KL}}(P||Q)$ denotes the Kullback–Leibler divergence between distributions $P$ and $Q$, and $\beta$ is a weighting coefficient. For variable-size crystal structures, we normalize the KL contribution according to the chosen structure-size normalization strategy to obtain a structure-level regularization term (see Appendix A). The regularization keeps the reinforced policy close to the reference policy $p^{\theta_{\text{ref}}}$ of the pretrained generative model, which is critical for preserving its learned inductive biases.

# 3. Inference-Time Reinforcement Learning

OMatG can be trained to predict only the velocity field $b^\theta(t, x_t)$ via Eq. (2), or to jointly predict the velocity field and the denoiser $z^\theta(t, x_t)$ via Eq. (5). When the denoiser—and thus the score—is available, policy-gradient RL can be applied directly to the resulting stochastic sampler (see Section 3.1). In contrast, when only the velocity field is learned and no explicit score is available at inference time, we add stochastic perturbations to the underlying ODE dynamics to define a surrogate stochastic process. By enabling exploration, policy-ratio estimation, and KL regularization without requiring access to the score, this process allows policy-gradient updates to the full pretrained OMatG model (see Section 3.2). Finally, using the same stochastic-perturbation idea, we apply policy-gradient RL to learn a time-dependent velocity-annealing schedule that rescales the frozen velocity field, replacing handcrafted annealing schemes (see Section 3.3).

## 3.1. Score-Based OMatG-IRL

The Euler–Maruyama updates in Eq. (7) based on $b^\theta(t, x_t)$ and $z^\theta(t, x_t)$ define a stochastic policy $\pi^\theta(a_t|s_t) = p^\theta(x_{t+\Delta t}|x_t)$, which makes the policy-gradient RL framework of Section 2.3 directly applicable. Compared to existing RL applications in (latent) diffusion (Park & Walsh, 2026) or flow-matching frameworks (Liu et al., 2025), policy updates here jointly modify both the velocity field $b^\theta(t, x_t)$ and the denoiser $z^\theta(t, x_t)$ through updates to the shared CSPNet backbone of OMatG, rather than acting on only a single component. In addition to the KL regularization in Eq. (11), we introduce an explicit denoiser-distillation regularization term:

$$\mathcal{J}_{\text{Dist}}(\theta) = -\frac{\delta}{GN_t} \sum_{i,t} \big\| z^\theta(t, x_t^i) - z^{\theta_{\text{ref}}}(t, x_t^i) \big\|_{\text{str}}^2. \tag{12}$$

Here, $\|\cdot\|_{\text{str}}^2$ denotes a structure-normalized squared norm. For atomic positions, this corresponds to summing over Cartesian components and averaging over atoms. Increasing the weight $\delta$ constrains the reinforced denoiser to remain close to the pretrained model. While the KL objective constrains the full drift distribution [which depends jointly on $b^\theta(t, x_t)$ and $z^\theta(t, x_t)$], the denoiser-distillation term acts directly on $z^\theta(t, x_t)$ and thus explicitly controls the score.

## 3.2. Velocity-Based OMatG-IRL

The Euler updates in Eq. (4) based on $b^\theta(t, x_t)$ are deterministic, which is problematic for policy-gradient RL: Policy-ratio computation typically requires expensive divergence estimation, and the absence of per-step stochasticity severely limits exploration (Liu et al., 2025). A natural remedy would be to convert the ODE into an SDE that preserves the same marginal distributions $\rho_t$ while providing the stochasticity required for RL (as done in Flow-GRPO for the specific case of linear interpolants with Gaussian base distributions; see Appendix B). Within the SI framework, this role is generally played by the SDE in Eq. (6) but in the setting of this section, the required denoiser $z^\theta(t, x_t)$ is not available.

Instead, we introduce a surrogate stochastic process by augmenting the dynamics with a noise schedule $\sigma_{\text{ref}}(t)$:

$$x_{t+\Delta t} = x_t + b^{\theta_{\text{ref}}}(t, x_t)\Delta t + \sigma_{\text{ref}}(t)\sqrt{\Delta t}\,\xi. \tag{13}$$

The omitted score correction is subdominant, $\mathcal{O}(\sigma_{\text{ref}}^2)$, so the resulting discrepancy is likewise bounded by $\mathcal{O}(\sigma_{\text{ref}}^2)$, as motivated by Girsanov's theorem (Øksendal, 2013) (see Appendix C for the corresponding discrete-time path-space KL argument). Empirically, evaluation metrics on the final samples confirm that the distribution remains virtually unchanged for sufficiently small $\sigma_{\text{ref}}(t)$ (see Section 5).

The stochastic surrogate dynamics both enable exploration and allow policy-gradient updates to the velocity field

$b^{\theta_{\mathrm{ref}}}(t, x_t) \to b^{\theta}(t, x_t)$ through updates to the weights of OMatG's CSPNet backbone (see Fig. 1). Importantly, the construction in Eq. (13) defines a well-posed reference policy for KL regularization. During RL, however, we are not restricted to the noise schedule $\sigma_{\mathrm{ref}}(t)$ and may use alternative schedules $\sigma(t)$ to enhance exploration.

### 3.3. Velocity-Annealing OMatG-IRL

It has been empirically observed that velocity annealing improves the performance of flow-matching and SI models (Yim et al., 2023; Bose et al., 2024; Miller et al., 2024; Höllmer et al., 2025). In this approach, the velocity is modified during generation as $b^{\theta}(t, x_t) \to (1 + st)\, b^{\theta}(t, x_t)$, where $s$ is a scalar hyperparameter.

In OMatG-IRL, we introduce a learned, time-dependent velocity-annealing schedule $s^{\theta}(t)$. The reference policy is given by

$$x_{t+\Delta t} = x_t + b^{\theta_{\mathrm{ref}}} \Delta t + \sigma_{\mathrm{ref}}(t)\, b^{\theta_{\mathrm{ref}}} \sqrt{\Delta t}\, \xi, \quad (14)$$

where $b^{\theta_{\mathrm{ref}}} = b^{\theta_{\mathrm{ref}}}(t, x_t)$ is the frozen velocity field of the pretrained generative model and $\xi \sim \mathcal{N}(0, 1)$ is a standard Gaussian random variable. Policy-gradient RL then reinforces a zero-initialized schedule $s^{\theta}(t)$ in the velocity-annealed update

$$x_{t+\Delta t} = x_t + [1 + s^{\theta}(t)] b^{\theta_{\mathrm{ref}}} \Delta t + \sigma(t)\, b^{\theta_{\mathrm{ref}}} \sqrt{\Delta t}\, \xi. \quad (15)$$

This procedure is akin to residual policy learning (Silver et al., 2019), in that we learn a residual correction to a frozen pretrained model using a separate lightweight network. Because $s^{\theta}(t)$ depends only on the one-dimensional time variable, we use a simple multilayer perceptron.

## 4. Datasets and Metrics

In this work, we reinforce OMatG models trained on the *MP-20* dataset, which contains $45\,229$ structures from the Materials Project with at most 20 atoms per unit cell, spanning a wide range of structures and compositions across 89 atomic species (Jain et al., 2013; Xie et al., 2022). All structures in the dataset are relaxed using density-functional theory and together comprise most experimentally known inorganic materials with up to 20 atoms per unit cell.

As performance metrics, we consider the *match rate*, *RMSE*, *relative energy per atom*, and *invalid-energy rate*. For the match rate (Xie et al., 2022), we generate one structure for each composition in a reference dataset not seen during training and compare the generated and reference structures using Pymatgen's `StructureMatcher` with tolerances `stol=0.5`, `ltol=0.3`, and `angle_tol=10.0` (Ong et al., 2013). We then report the fraction of successful matches. The RMSE is computed as the average root-mean-square displacement between matched structures, normalized by $(V/N)^{1/3}$, where $V$ is the matched volume and $N$

is the number of atoms (Xie et al., 2022). The energy per atom of generated structures is evaluated using the MACE-MPA-0 model (Batatia et al., 2025), and we report energies relative to the reference structure of the same composition. Since the predicted energy can diverge for unphysical structures (e.g., due to atomic overlaps; see Appendix D), such structures are excluded from the energy statistics. We report the fraction of these samples as the invalid-energy rate.

We additionally consider an OMatG model trained on the *MP-20-polymorph-split* dataset (Martirossyan et al., 2025), a re-splitting of MP-20 in which polymorphs (i.e., distinct crystal structures sharing the same composition) are grouped into the same split. For this setting, we evaluate performance using *METRe* and *cRMSE* instead of match rate and RMSE. METRe measures how well generated structures cover the reference dataset by identifying the best match for each reference structure, while cRMSE reports the average normalized root-mean-square displacement between best-matching pairs, with unmatched reference structures penalized using `stol`.

Although the fraction of compositions with multiple polymorphs in MP-20 is relatively modest [$\sim 82\,\%$ of compositions are unique (Martirossyan et al., 2025)], evaluation on the polymorph split using METRe and cRMSE provides a more principled assessment of the generative model. Match rate and RMSE still remain strongly correlated with METRe and cRMSE on this dataset, so we also report results on the standard MP-20 split to enable direct comparison with previously published OMatG models. Finally, because the energy differences between polymorphs are small (up to $\sim 0.1\,\mathrm{eV}$ per atom), the choice of split does not materially affect the reported relative energies per atom.

## 5. Experiments

We use OMatG-IRL to present the first application of policy-gradient RL to CSP. We show that the relative energy per atom can be effectively reinforced, achieving reductions of approximately $0.5\,\mathrm{eV}$ per atom (see Section 5.1). The flexible design of OMatG further allows us to demonstrate that the velocity-only RL approach of Section 3.2 achieves robust performance comparable to the score-based RL approach in Section 3.1. Finally, we show that velocity-annealing OMatG-IRL can replace handcrafted annealing schedules while enabling accurate CSP with at least one order of magnitude fewer integration steps (see Section 5.2).

### 5.1. Inference-Time Energy Reinforcement

Before presenting the RL performance of score-based and velocity-based OMatG-IRL in Section 5.1.2, we first motivate the relative energy per atom as a transferable reinforcement signal for CSP in Section 5.1.1.

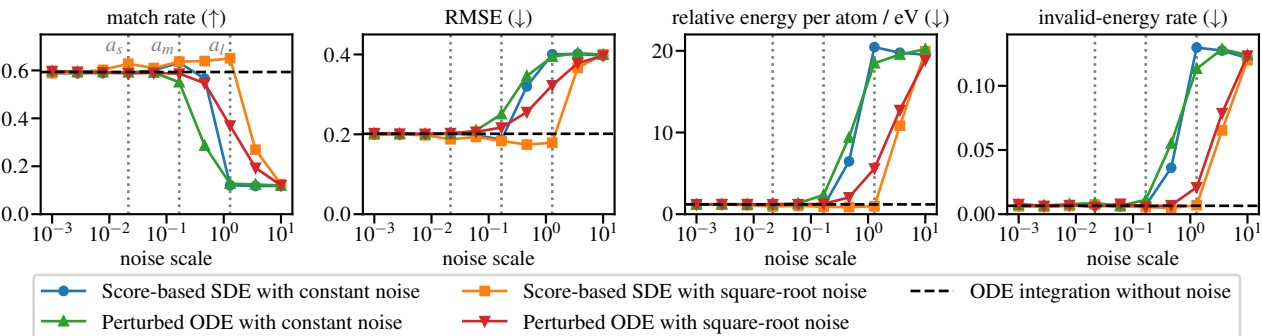

*Figure 2.* Test-set evaluation metrics for score-based SDE and perturbed velocity-based ODE integration of the atomic positions under different noise schedules. Small, medium, and large noise scales are denoted by $a_s$, $a_m$, and $a_l$, respectively.

### 5.1.1. ENERGY-BASED OBJECTIVE

The CSP task is arguably harder to reinforce than the DNG task. In DNG, if the species-generation process is also reinforced, a generative model can quickly focus on generating specific compositions that optimize the reward, effectively "hacking" the objective. The existing approaches in MatInvent (Chen et al., 2025) and Chemeleon2 (Park & Walsh, 2026) explicitly counteract this behavior by introducing diversity rewards. In the CSP setting considered here, the composition is fixed throughout generation, and diversity is naturally enforced through composition conditioning.

Common structure-based CSP metrics such as match rate, METRe, or cRMSE are ill-suited as reinforcement signals. These metrics measure similarity to specific ground-truth structures in the training set, an objective that is closely aligned with the original training signal. As a result, the additional learning signal by RL would be limited beyond what is already captured during pretraining. Furthermore, these metrics are inherently composition-dependent as each composition corresponds to a different set of target structures. This leads to sparse and poorly transferable rewards that do not generalize well across compositions. Optimizing such rewards would require repeated sweeps over large portions of the training dataset, making this approach computationally prohibitive.

In contrast, the relative energy per atom provides a physically meaningful and transferable reinforcement objective. The chemical rules that govern energy minimization (such as removing atomic overlaps) generalize across compositions. Reinforcing energy therefore promotes chemically reasonable structures independent of the specific target structure for a given composition. It also renders explicit a requirement that is already implicit in structure-based metrics, which assess stability only by comparing generated structures to known stable references. Energy-based reinforcement directly optimizes this underlying notion of chemical (meta-)stability, making it a natural and effective choice for RL in the CSP task.

### 5.1.2. SCORE- VS. VELOCITY-BASED OMATG-IRL

To compare the score-based OMatG-IRL approach of Section 3.1 to the velocity-based OMatG-IRL approach of Section 3.2, we use the publicly available pretrained Trig-SDE-Gamma model of OMatG,[1] which was trained for the CSP task on the MP-20 dataset (Höllmer et al., 2025). For the atomic positions, this model predicts both the velocity field $b^\theta(t, x_t)$ and the denoiser $z^\theta(t, x_t)$ and can be integrated either in SDE mode via Eq. (7) or in perturbed ODE mode via Eq. (13). All modifications discussed below apply exclusively to the atomic positions, which constitute the harder-to-learn component of the CSP task. The lattice vectors are always integrated with an unperturbed ODE via Eq. (4) based on the frozen velocity field $b^{\theta_\text{ref}}(t, x_t)$.

In its original configuration, the Trig-SDE-Gamma OMatG model achieves a competitive match rate of approximately $69\%$ but relies on strong velocity annealing and a large number $N_t = 740$ of integration steps. For OMatG-IRL, we reduce the number of integration steps to $N_t = 50$ to limit computational cost and reduce variance in the terminal rewards $r^i = r(x_{t=1}^i)$. We also remove velocity annealing in the pretrained model, as we find that keeping annealing while reducing the number of integration steps leads to severe performance degradation, indicating that annealing and time discretization are tightly coupled in the pretrained model (see Appendix E).

We consider two noise schedules for stochastic integration: a constant schedule $\sigma(t) = a$, and, inspired by Flow-GRPO, a square-root schedule $\sigma(t) = a\sqrt{(1-t)/t}$ which decays to zero as $t \to 1$. For both schedules, increasing the noise scale $a$ beyond a moderate range deteriorates the performance of the pretrained model (see Fig. 2). Score-based SDE integration tolerates substantially larger noise levels before performance degrades due to the explicit drift correction provided by the learned score. In contrast, perturbed ODE integration is more sensitive to noise and requires

---

[1]https://huggingface.co/OMatG/MP-20-CSP

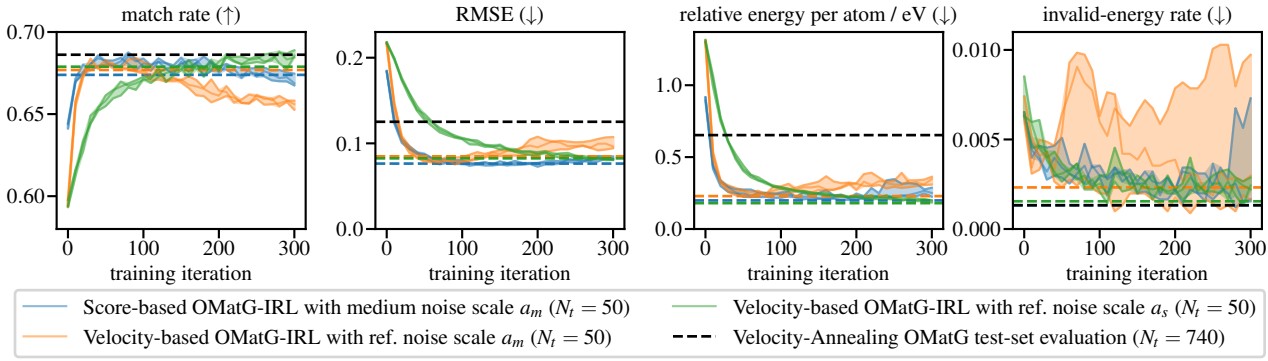

*Figure 3.* Evolution of validation metrics for score-based and velocity-based OMatG-IRL as a function of RL training iteration, shown for three random seeds of the same setup ($N_t = 50$). The colored dashed lines indicate the test-set performance of the OMatG-IRL checkpoint selected by the validation optimum. For reference, we also show the test-set performance of the original velocity-annealed OMatG model evaluated with $N_t = 740$ integration steps. The velocity-based OMatG-IRL setup with small reference noise scale $a_s$ (green) uses fewer PPO epochs per update, resulting in slower but more stable reinforcement.

smaller perturbations to preserve baseline performance. Independently of the integration mode, the square-root noise schedule remains stable for larger noise scales $a$ compared to the constant schedule.

Based on these observations, we use the square-root noise schedule for all policy-gradient RL experiments. For velocity-based OMatG-IRL, we compare small and medium reference noise scales ($a_{\mathrm{ref}} = a_s$ and $a_{\mathrm{ref}} = a_m$) to define the KL-regularized reference policy, and use the medium noise scale $a_m$ during policy rollouts for exploration (see Fig. 2). For score-based OMatG-IRL, we use the medium noise scale $a_m$, which yields the best RL performance and thus ensures a fair comparison with the velocity-based approach (see Appendix F for a detailed noise-scale sensitivity analysis of both variants). As the reward, we use the negative energy per atom, which is equivalent to using the negative *relative* energy per atom because all structures within a GRPO group share the same composition and constant energy offsets cancel in the group-relative advantages. Invalid structures are assigned a fixed penalty energy, and the resulting energies are optionally clipped to reduce the influence of outliers on the group-normalized advantages and improve the stability of policy updates (see Appendix G).

We run a hyperparameter sweep on the validation reward to determine the optimal configurations for both score- and velocity-based OMatG-IRL (see Appendix H). We find that both approaches are able to effectively reinforce the relative energy per atom (see Fig. 3). At their respective validation optima, both methods achieve relative energies that are approximately $0.5\,\mathrm{eV}$ per atom lower than the pretrained Trig-SDE-Gamma OMatG model evaluated with optimal velocity annealing and $N_t = 740$ integration steps. Despite operating with only $N_t = 50$ integration steps, energy reinforcement also reduces the RMSE and maintains the match rate. These results demonstrate that energy-based inference-

time RL can significantly improve CSP performance while reducing the computational cost of generation by more than an order of magnitude. After post-generation relaxation, the pretrained OMatG model and OMatG-IRL reach similar final CSP performance with even lower RMSE values and relative energies per atom (see Appendix I). However, generated structures from OMatG-IRL require fewer relaxation steps, indicating that energy-based reinforcement partially amortizes the computationally expensive relaxation process.

Importantly, the velocity-based OMatG-IRL approach matches the performance of the score-based variant, showing that effective policy-gradient RL is possible without access to an explicit score at inference time and thus applicable to a broader class of flow-based generative models. Additional robustness experiments in Appendix J show that, using the same RL setup and hyperparameters, velocity-based OMatG-IRL consistently improves RMSE across multiple CSP datasets and pretrained OMatG models while using far fewer integration steps. Finally, we demonstrate the flexibility of velocity-based OMatG-IRL in Appendix K, where it is applied to hybrid DNG experiments in which the discrete species-generation process is left unchanged and only the continuous atomic-position updates are reinforced. In this setting, OMatG-IRL can optimize both energy-based and non-differentiable symmetry-based rewards.

### 5.2. Inference-Time Velocity-Annealing Reinforcement

We apply the velocity-annealing variant of OMatG-IRL introduced in Section 3.3 to the best-performing CSP model of OMatG, namely the Linear-ODE model trained on the MP-20-polymorph-split dataset (Martirossyan et al., 2025). This pretrained model achieves a METRe score of $\sim 70\,\%$ and a cRMSE of $\sim 0.19$, but only when using a large number of integration steps ($N_t = 950$) together with carefully tuned velocity annealing on atomic positions and lattice vectors.

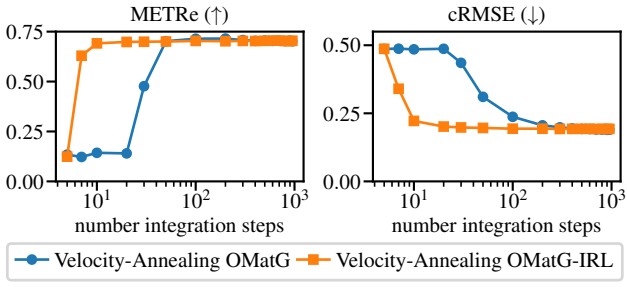

*Figure 4.* Test-set evaluation metrics for velocity-annealing OMatG and OMatG-IRL as a function of the number of integration steps $N_t$, highlighting the improved robustness of OMatG-IRL to aggressive time discretization.

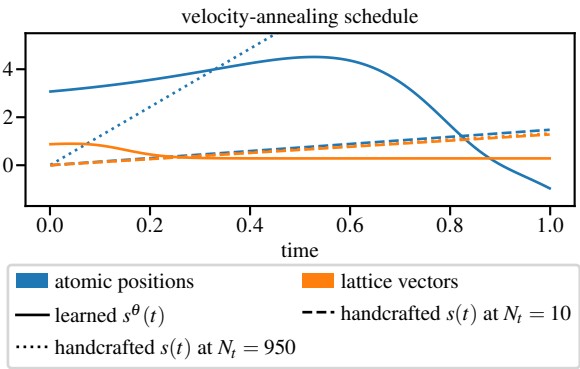

*Figure 5.* Velocity-annealing schedules for the atomic positions and lattice vectors, either learned with OMatG-IRL or obtained from a hyperparameter sweep over a handcrafted schedule at a given number of integration steps. The learned schedules adapt qualitatively differently from handcrafted ones.

Similar to Section 5.1, we remove the handcrafted velocity annealing from the pretrained model and reduce the number of integration steps to $N_t = 100$ for velocity-annealing OMatG-IRL. We first identify an appropriate (constant) reference noise schedule $\sigma_{\text{ref}}(t)$ in Eq. (14), and then apply policy-gradient RL to learn a time-dependent velocity-annealing schedule $s^\theta(t)$ for both atomic positions and lattice vectors. Because the learned annealing schedule depends only on time (and not on composition), and because velocity annealing is known to directly improve structural metrics, we can reinforce the model using a cRMSE-like objective in this setting (see Appendix L for details).

Despite using only $N_t = 100$ integration steps, velocity-annealing OMatG-IRL fully recovers the performance of the original OMatG model that requires $N_t = 950$ steps. More remarkably, the learned velocity-annealing schedule $s^\theta(t)$ yields robust performance across a wide range of integration step counts (see Fig. 4), allowing the number of steps to be reduced as far as $N_t = 10$ with only minor degradation. This stands in sharp contrast to the original OMatG model, whose performance rapidly deteriorates unless the number of integration steps is at least an order of magnitude larger.

We verified that a hyperparameter sweep over the handcrafted velocity-annealing schedule at the low integration step count of $N_t = 10$ does not recover the performance achieved by the learned velocity-annealing schedule, indicating that the improvement indeed arises from the learned annealing policy (see Appendix M). For the lattice vectors, the learned schedule $s^\theta(t)$ quickly drops to a constant value, whereas the handcrafted schedule is constrained to increase monotonically in time with a slope that appears largely independent of the number of integration steps $N_t$. For the atomic positions, in contrast, the slope of the handcrafted schedule depends strongly on $N_t$, while the learned schedule initially increases but then decreases after $t \approx 0.5$, eventually becoming negative. Importantly, $1 + s^\theta(t)$ remains positive at all times, so that the effective velocity $[1 + s^\theta(t)]\, b^\theta(t, x_t)$ is never reversed (see Fig. 5).

## 6. Conclusion

We introduced OMatG-IRL, a policy-gradient RL framework for continuous-time generative models of crystalline materials, and demonstrated its effectiveness on CSP. A key result is that effective policy-gradient RL is possible even in velocity-only flow-based models, without access to an explicit score. This makes the framework applicable beyond OMatG to other continuous-time flow-based generative models whenever suitable task-specific rewards are available. Our results further show that velocity-annealing schedules can be learned through policy-gradient RL, substantially reducing the required number of integration steps.

Despite these promising results, several limitations remain. First, velocity-based OMatG-IRL uses a surrogate stochastic process whose marginals deviate, albeit in a controlled manner, from those of the original ODE and the corresponding score-based SDE. This limits the accessible exploration noise compared to score-based OMatG-IRL and may make the optimal noise scale inaccessible in other settings. Second, our energy-based reward improves the relative energy per atom and RMSE, but does not directly optimize structure-matching metrics such as match rate or METRe. More broadly, comparing OMatG-IRL to energy-based guidance methods may help clarify the trade-offs between black-box policy-gradient reinforcement and differentiable test-time guidance. Third, fully extending OMatG-IRL to DNG within the OMatG framework—including reinforcement of the species-generation process—requires addressing the discrete flow-matching implementation for the atomic species (Campbell et al., 2024). Conversely, our results suggest that highly accurate, diverse, and reinforced CSP models, when combined with a separate composition predictor, could also serve as powerful building blocks for scalable DNG in inverse materials design pipelines.

## Acknowledgements

The authors acknowledge funding from NSF Grant OAC-2311632. This work was also supported by a grant from the Simons Foundation [MPS-T-MPS-00839534, MET].

We gratefully acknowledge use of the research computing resources of the Empire AI Consortium, Inc, with support from Empire State Development of the State of New York, the Simons Foundation, and the Secunda Family Foundation (Bloom et al., 2025). Moreover, the authors gratefully acknowledge the additional computational resources and consultation support provided by the IT High Performance Computing at New York University.

## Impact Statement

This paper presents work whose goal is to improve targeted generative modeling for crystalline materials, which could accelerate the discovery of materials relevant to clean energy, catalysis, electronics, and other technologies. More efficient crystal generation may also reduce the computational cost of materials screening. At the same time, methods for targeted materials design may carry risks if applied irresponsibly, for example toward the design of materials with harmful environmental, safety, or security implications. In addition, large-scale model training and screening consume computational resources and therefore carry an environmental cost.

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

## A. Normalization for Variable-Size Structures

Crystalline-material datasets contain structures with varying numbers of atoms, which causes both the GRPO objective in Eq. (10) and the KL regularization in Eq. (11) to scale with structure size. Without correction, this would bias policy updates toward larger structures. To prevent such size-dependent effects, we can either divide the advantages by the number of atoms in the structure, or we compute PPO policy ratios per atom and average them to obtain a structure-level policy loss. For the KL regularization, we can similarly average the per-atom KL divergence to obtain a structure-level penalty. The choice of normalization strategy is treated as a hyperparameter and part of our hyperparameter sweep, which also includes the option of applying no explicit structure-size normalization (see Appendix H).

## B. Comparison to Flow-GRPO

Flow-GRPO provides an alternative route for applying policy-gradient RL to flow-based generative models that only learn a velocity $b^\theta(t, x_t)$ (Liu et al., 2025). Its construction relies on the fact that, for linear interpolants $x_t = \alpha(t)\,x_0 + \beta(t)\,x_1$ with Gaussian base distribution $x_0 \sim \mathcal{N}(0, \boldsymbol{I})$, the marginal score can be expressed in terms of the velocity field $b^\theta(t, x_t)$ (Albergo et al., 2025; Domingo-Enrich et al., 2025). This allows one to convert the generative ODE in Eq. (3) directly into an equivalent, marginal-preserving SDE of the form in Eq. (6), while still depending only on $b^\theta(t, x_t)$. Specifically, with the "over-dot" notation denoting time derivatives, the Euler–Maruyama updates with step size $\Delta t$ become (Albergo et al., 2025; Domingo-Enrich et al., 2025; Liu et al., 2025)

$$x_{t+\Delta t} = x_t + \left\{ b^\theta(t, x_t) + \frac{\sigma^2(t)}{2\alpha(t)\left[\alpha(t)\dot\beta(t)/\beta(t) - \dot\alpha(t)\right]} \left[ b^\theta(t, x_t) - \frac{\dot\beta(t)}{\beta(t)} x_t \right] \right\} \Delta t + \sigma(t)\sqrt{\Delta t}\,\xi. \qquad (16)$$

These updates define a stochastic policy and enable policy-gradient RL with tractable policy ratios and KL regularization. However, Flow-GRPO relies on an analytic score–velocity relation, which is only available in the setting of linear interpolants with Gaussian base distributions.

Velocity-based OMatG-IRL does not attempt to recover the exact score and instead introduces the perturbative stochastic surrogate in Eq. (13). This makes the approach applicable across the full SI design space used by OMatG, including, in particular, settings with arbitrary base distributions and periodic fractional-coordinate domains. The resulting trade-off is that the surrogate process is not exactly marginal-preserving at nonzero noise scale, but the discrepancy is bounded by terms of order $\mathcal{O}(\sigma^2)$ and evaluation metrics remain virtually unchanged for sufficiently small $\sigma(t)$ (see Section 5 and Appendix C).

## C. Validation of the Surrogate Stochastic Process

In Section 3.2, we introduced the surrogate noisy Euler updates in Eq. (13) that omit the score-correction term $-\sigma_{\mathrm{ref}}^2(t)z^\theta(t, x_t)/[2\gamma(t)]$ from the drift of the score-based Euler–Maruyama update in Eq. (6). While the continuous-time change of measure between the path laws induced by the corresponding SDEs can be motivated by Girsanov's theorem, our implementation is discrete-time. We therefore quantify the discrepancy directly at the level of the discretized transition kernels and their induced path measures.

The one-step conditional kernels $p_{\mathrm{score}}(x_{t+\Delta t}|x_t)$ and $p_{\mathrm{surr}}(x_{t+\Delta t}|x_t)$ of the score-based and surrogate updates are both Gaussian distributions with the same covariance, differing only in their means. Their conditional KL divergence is therefore

$$D_{\mathrm{KL}}\big[p_{\mathrm{score}}(x_{t+\Delta t}|x_t) \,\|\, p_{\mathrm{surr}}(x_{t+\Delta t}|x_t)\big] = \frac{\sigma_{\mathrm{ref}}^2(t)}{8\gamma^2(t)}\|z^\theta(t, x_t)\|^2 \Delta t. \qquad (17)$$

Since both discretized processes define Markov chains with common initial distribution $\rho_0$, their path measures factorize as $\mathcal{P}_{\mathrm{score}} = \rho_0(x_0)\prod_t p_{\mathrm{score}}(x_{t+\Delta t}|x_t)$ and $\mathcal{P}_{\mathrm{surr}} = \rho_0(x_0)\prod_t p_{\mathrm{surr}}(x_{t+\Delta t}|x_t)$, and the KL divergence $D_{\mathrm{KL}}(\mathcal{P}_{\mathrm{score}}\|\mathcal{P}_{\mathrm{surr}})$ decomposes into a sum of expected conditional KL terms (Cover & Thomas, 1991). Assuming the same initial distribution $\rho_0(x_0)$,

$$D_{\mathrm{KL}}(\mathcal{P}_{\mathrm{score}}\|\mathcal{P}_{\mathrm{surr}}) = \sum_t \mathbb{E}_{\mathcal{P}_{\mathrm{score}}}\Big\{ D_{\mathrm{KL}}\big[p_{\mathrm{score}}(\cdot|X_t) \,\|\, p_{\mathrm{surr}}(\cdot|X_t)\big]\Big\}. \qquad (18)$$

We estimate this forward-path KL by integrating the score-based dynamics and accumulating the per-step contributions along the visited states. The reverse-path KL $D_{\mathrm{KL}}(\mathcal{P}_{\mathrm{surr}}\|\mathcal{P}_{\mathrm{score}})$ is estimated analogously using trajectories from the surrogate

dynamics. These path-space KL divergences upper-bound the KL divergence between the corresponding terminal marginals, and therefore provide a direct measure of how strongly the surrogate stochastic dynamics can perturb the final sample distribution. Moreover, Eqs. (17) and (18) imply an $\mathcal{O}(\sigma_{\text{ref}}^2)$ scaling of this discrepancy for well-behaved $z^\theta(t, x_t)$ and $\gamma(t)$.

To verify the predicted $\mathcal{O}(\sigma_{\text{ref}}^2)$ scaling in the setup of Section 5.1, we integrate both the score-based and surrogate SDEs of the pretrained Trig-SDE-Gamma OMatG model with $N_t = 50$ integration steps using the square-root noise schedule $\sigma_{\text{ref}}(t) = a_{\text{ref}}\sqrt{(1 - t)/t}$ at various noise scales $a_{\text{ref}}$. We then estimate both forward- and reverse-path KL divergences. To make structures with different numbers $N$ of atoms comparable, we normalize each path-space KL by $N$ and average the resulting per-atom values over structures. The path-space KL divergences in both directions exhibit the expected $a_{\text{ref}}^2$ scaling and remain similar across all considered noise scales (see Fig. 6), indicating that the score-based and surrogate processes remain close at the level of path laws.

In addition, we report the per-atom ratio between the norm of the score-correction term $-\sigma_{\text{ref}}^2(t)z^\theta(t, x_t)/[2\gamma(t)]$ and the norm of the realized one-step displacement $x_{t+\Delta t} - x_t$, averaged over time steps during numerical integration (see Fig. 6). This score-correction ratio provides a more interpretable measure of the size of the score-correction term relative to the actual one-step motion induced by the sampler. For small noise scales, the score-correction ratio remains small in both directions, indicating that the score-correction term is negligible relative to the realized one-step motion in the perturbative regime. The score-correction ratio then grows approximately linearly with $a_{\text{ref}}$, consistent with the score-correction term scaling as $\sigma_{\text{ref}}^2$ while the realized one-step displacement contains a stochastic increment scaling as $\sigma_{\text{ref}}$. The increasing asymmetry between the forward and reverse score-correction ratios at larger $a_{\text{ref}}$ reflects the fact that this ratio is a local, state-dependent diagnostic and is therefore more sensitive than the path-space KL to differences in the states visited by the two processes. The larger reverse score-correction ratio beyond moderate noise scales indicates that surrogate-generated trajectories increasingly visit states where the score correction omitted by the surrogate is more significant relative to the realized one-step motion, which is consistent with the earlier deterioration of evaluation metrics for the final structures in the high-noise regime (see Fig. 2).

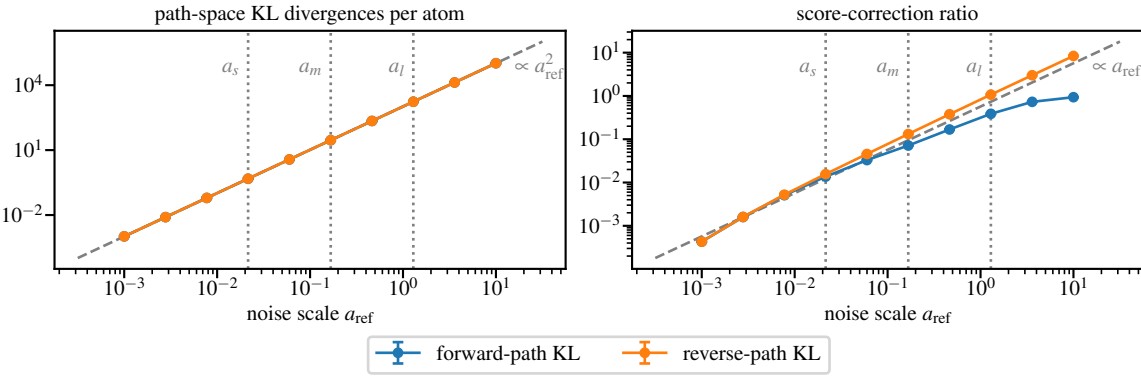

*Figure 6.* Path-space KL and score-correction diagnostics for the score-based and surrogate stochastic processes of the pretrained Trig-SDE-Gamma OMatG model under the square-root noise schedule $\sigma_{\text{ref}}(t) = a_{\text{ref}}\sqrt{(1 - t)/t}$, averaged over 2048 structures and computed from trajectories with $N_t = 50$ integration steps. The left plot shows forward- and reverse-path KL divergences per atom as a function of $a_{\text{ref}}$. The right plot shows forward and reverse score-correction ratios. Error bars indicate standard deviations over three random sets of initial configurations.

# D. Invalid-Energy Rate

We exclude invalid generated structures from the energy statistics and instead report the fraction of such invalid samples. In OMatG-IRL, a generated structure is considered invalid if it satisfies at least one of the following criteria:

1. The unit-cell volume is smaller than $0.1\,\text{Å}^3$.

2. Any pairwise atomic distance (accounting for periodic boundary conditions) is smaller than $0.5\,\text{Å}$.

3. The polar sine of the lattice vectors is smaller than $10^{-3}$, indicating a nearly degenerate unit cell.

## E. Sensitivity of OMatG to Integration Steps and Velocity Annealing

In Table 1, we show that the performance of the pretrained Trig-SDE-Gamma model in OMatG is largely unchanged when switching between SDE integration via Eq. (7) and ODE integration via Eq. (4). However, when the number of integration steps is reduced while retaining velocity annealing, performance deteriorates severely. This effect is illustrated for the ODE-based integration, where no additional noise schedule needs to be tuned. In contrast, when velocity annealing is removed, reducing the number of integration steps has only a minor impact on performance, indicating that velocity annealing and number of integration steps are tightly coupled in the pretrained model.

*Table 1.* Match rate and RMSE of the Trig-SDE-Gamma OMatG model for different integration modes, velocity-annealing settings, and integration step counts, illustrating the strong coupling between annealing and step number.

| Integration type | Velocity annealing | Integration steps | Match rate (%) ↑ | RMSE ↓ |
|---|---|---|---|---|
| SDE | ✓ | 740 | 68.62 | 0.1252 |
| ODE | ✓ | 740 | 67.30 | 0.1145 |
| ODE | ✓ | 100 | 55.07 | 0.3644 |
| ODE | ✓ | 50 | 13.39 | 0.3932 |
| ODE | ✗ | 740 | 60.59 | 0.1749 |
| ODE | ✗ | 100 | 60.05 | 0.1891 |
| ODE | ✗ | 50 | 59.40 | 0.2024 |

## F. OMatG-IRL at Different Noise Scales

In this section, we explore the influence of the noise scale on score-based OMatG-IRL (see Appendix F.1) and velocity-based OMatG-IRL (see Appendix F.2). We follow the setup of Section 5.1 and report the evolution of validation metrics during RL training together with the corresponding test-set evaluations of the OMatG-IRL checkpoints selected by the validation optima (analogous to Fig. 3).

### F.1. Score-Based OMatG-IRL

In Fig. 7, we compare the score-based variant of OMatG-IRL under the three noise scales $a_s$, $a_m$, and $a_l$ of the square-root noise schedule highlighted in Fig. 2. The medium noise scale $a_m$ yields the most effective reinforcement behavior: The validation relative energy per atom decreases rapidly and reaches the lowest value among the three noise scales. The small noise scale $a_s$ leads to slower improvement due to insufficient exploration, while the large noise scale $a_l$ is counterproductive for energy minimization. The corresponding test-set evaluations are consistent with the validation trends, and the checkpoint selected under the medium noise scale $a_m$ achieves the strongest relative-energy and RMSE improvements. This dependence on the noise scale is consistent with observations reported in Flow-GRPO (Liu et al., 2025). Based on these results, we use the score-based OMatG-IRL model with medium noise scale $a_m$ for comparison with the velocity-only approach in the main experiments in Section 5.1.

### F.2. Velocity-Based OMatG-IRL

For velocity-based OMatG-IRL, we explore the effect of using different noise scales in the square-root schedule for policy rollouts and KL regularization, respectively (see Section 3.2). Specifically, we distinguish between the exploration noise scale $a_{\text{exp}}$ used during policy rollouts and the reference noise scale $a_{\text{ref}}$ used to define the KL-regularized reference policy. In Fig. 8, we first fix the reference policy at the medium noise scale, $a_{\text{ref}} = a_m$, and vary the exploration scale over the three noise scales defined in Fig. 2, $a_{\text{exp}} \in \{a_s, a_m, a_l\}$. As in the score-based case, the medium exploration scale $a_{\text{exp}} = a_m$ provides the most effective reinforcement, whereas $a_{\text{exp}} = a_s$ leads to slower improvement due to insufficient exploration and $a_{\text{exp}} = a_l$ degrades energy-based reinforcement. The corresponding test-set evaluations are consistent with these validation trends and show the strongest relative-energy and RMSE improvements for the checkpoint selected at $a_{\text{exp}} = a_m$. In Fig. 9, we instead fix the exploration scale at $a_{\text{exp}} = a_m$ and vary the reference scale $a_{\text{ref}}$. In contrast to the strong dependence on the exploration noise $a_{\text{exp}}$, the choice of the reference noise $a_{\text{ref}}$ is far less influential and has little effect on the final test-set performance.

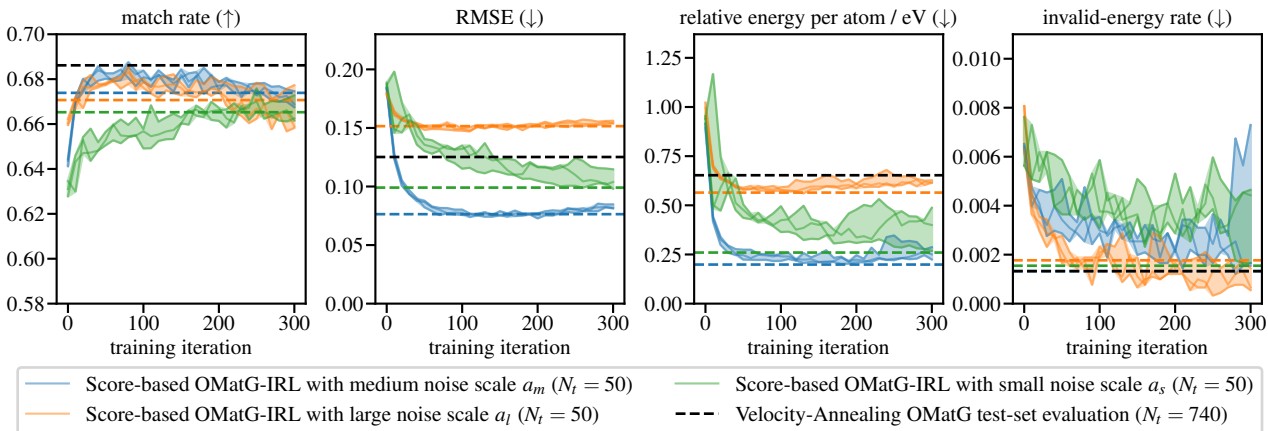

*Figure 7.* Evolution of validation metrics for score-based OMatG-IRL with different noise scales as a function of RL training iteration, shown for three random seeds of the same setup ($N_t = 50$). The colored dashed lines indicate the test-set performance of the OMatG-IRL checkpoint selected by the validation optimum. For reference, we also show the test-set performance of the original velocity-annealed OMatG model evaluated with $N_t = 740$ integration steps. The blue curve in this figure is identical to the blue curve shown in Fig. 3.

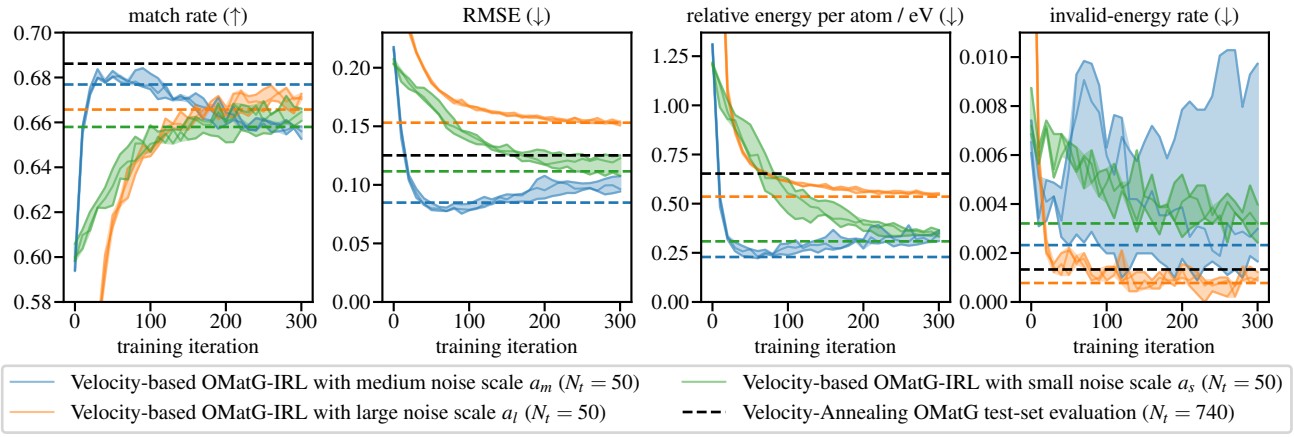

*Figure 8.* Evolution of validation metrics for velocity-based OMatG-IRL with different exploration noise scales $a_{\exp}$ at fixed reference noise scale $a_{\mathrm{ref}} = a_m$ as a function of RL training iteration, shown for three random seeds of the same setup ($N_t = 50$). The colored dashed lines indicate the test-set performance of the OMatG-IRL checkpoint selected by the validation optimum. For reference, we also show the test-set performance of the original velocity-annealed OMatG model evaluated with $N_t = 740$ integration steps. The blue curve in this figure is identical to the orange curve shown in Fig. 3.

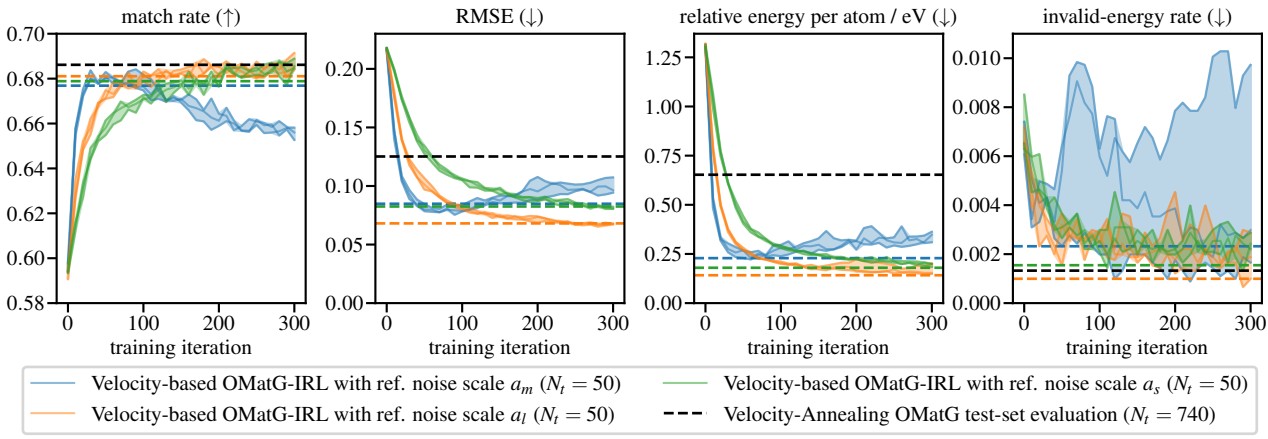

*Figure 9.* Evolution of validation metrics for velocity-based OMatG-IRL with different reference noise scales $a_{\text{ref}}$ at fixed exploration noise scale $a_{\text{exp}} = a_m$ as a function of RL training iteration, shown for three random seeds of the same setup ($N_t = 50$). The colored dashed lines indicate the test-set performance of the OMatG-IRL checkpoint selected by the validation optimum. For reference, we also show the test-set performance of the original velocity-annealed OMatG model evaluated with $N_t = 740$ integration steps. The blue curve in this figure is identical to the orange curve shown in Fig. 3. Also, the green curve in this figure is identical to the green curve in Fig. 3.

## G. Energy-Based Reward

As the energy-based rewards for a GRPO group of structures in OMatG-IRL, we first identify invalid structures (see Appendix D) and assign them a penalty energy of $3\,\text{eV}$ per atom, noting that typical energies per atom in the MP-20 dataset are well below $0\,\text{eV}$. For all remaining valid structures, we estimate the energy per atom using MACE-MPA-0 (Batatia et al., 2025) and optionally clip the resulting values to a band of three standard deviations around the group mean to prevent outliers from dominating the advantages. The reward is then defined as the negative of the (clipped) energy per atom, such that lower energies correspond to higher rewards. Since all structures within a GRPO group share the same composition, we do not need to compute explicit relative energies with respect to a reference structure. Any constant energy offset cancels out in the computation of group-relative advantages in Eq. (9).

## H. Hyperparameter Optimization

We optimize the hyperparameters of the policy-gradient RL setup in OMatG-IRL using Ray Tune (Liaw et al., 2018) with Optuna (Akiba et al., 2019) as the underlying sampler. For the RL experiments in Section 5.1, which adjust the velocity field $b^\theta(t, x_t)$ of the atomic positions, we tune the following hyperparameters (when applicable), sampled from the distributions below:

- GRPO group size $G \sim \text{Choice}(32, 64)$ (with a fixed number of 16 groups).

- Sharing initial positions within a GRPO group $\sim \text{Choice}(\text{True}, \text{False})$.

- Number of PPO epochs $\sim \text{Choice}(1, 2, 3, 4)$.

- PPO clipping parameter $\varepsilon \sim \text{Uniform}(0.05, 0.3)$.

- Policy loss weight $\sim \text{LogUniform}(0.1, 10.0)$.

- KL regularization weight $\sim \text{Choice}(0.0, 0.00001, 0.00003, 0.0001, 0.0003, 0.001, 0.003, 0.01)$.

- Denoiser-distillation weight $\sim \text{Choice}(0.0, 0.000001, 0.000003, 0.00001, 0.00003, 0.0001, 0.0003, 0.001)$.

- Adam learning rate $\sim \text{LogUniform}(0.00001, 0.001)$.

- Choice of normalization strategy for variable-size structures (see Appendix A).

- Energy clipping $\sim \text{Choice}(\text{True}, \text{False})$.

For the experiments in Section 5.2, which learn time-dependent scaling functions $s^\theta(t, x_t)$ for both positions and lattice parameters, we use a more aggressive search space (in particular, without any regularization):

- GRPO group size $G \sim \text{Choice}(32, 64)$ (with a fixed number of 16 groups).

- Sharing initial positions within a GRPO group $\sim \text{Choice}(\text{True}, \text{False})$.

- Number of PPO epochs $\sim \text{Choice}(1, 2, 3, 4, 5)$.

- PPO clipping parameter $\varepsilon \sim \text{Uniform}(0.1, 0.5)$.

- Position-policy loss weight $\sim \text{LogUniform}(0.01, 100.0)$.

- Adam learning rate $\sim \text{LogUniform}(0.00001, 0.01)$.

- Hidden dimension of the multilayer perceptron of $s^\theta(t) \sim \text{Choice}(32, 64, 128)$.

- Number of hidden layers of the multilayer perceptron of $s^\theta(t) \sim \text{Choice}(1, 2, 3)$.

- Shared trunk for the atomic-position and lattice-vector scale prediction in $s^\theta(t) \sim \text{Choice}(\text{True}, \text{False})$.

The policy weight of the lattice component is fixed to $1.0$. Only in this case, the position and lattice weights are rescaled to sum to one.

All hyperparameter optimization runs were conducted under a fixed computational budget, and final configurations were selected based on the validation reward.

## I. Post-Generation Relaxation

*Table 2.* Test-set evaluation metrics after relaxation for structures generated by the original velocity-annealed OMatG model and the reinforced OMatG-IRL model with small reference noise scale $a_{\text{ref}} = a_s$ (see the black dashed line and green curve in Fig. 3).

| Method | match rate ↑ | RMSE ↓ | relative energy per atom / eV ↓ | invalid-energy rate ↓ |
|---|---|---|---|---|
| Velocity-based OMatG-IRL with ref. noise scale $a_s$ ($N_t = 50$) | 67.09 % | 0.0582 | 0.0187 | 0 / 9046 |
| Velocity-Annealing OMatG ($N_t = 740$) | 68.06 % | 0.0536 | 0.0149 | 0 / 9046 |

For the setup of Section 5.1, we further assess the practical relevance of energy-based reinforcement by relaxing structures generated with the pretrained Trig-SDE-Gamma OMatG model and a reinforced OMatG-IRL variant. For relaxation, we use GPU-based LBFGS as implemented in TorchSim (Cohen et al., 2025), with energies, atomic forces, and unit-cell stresses computed by the MACE-MPA-0 machine-learned interatomic potential (Batatia et al., 2025). We jointly optimize atomic positions and lattice vectors for at most $1000$ steps, stopping earlier if the maximum force falls below $0.02\,\text{eV}/\text{Å}$. Relaxation leaves the match rate virtually unchanged but substantially reduces the RMSE, relative energy per atom, and invalid-energy rate. Importantly, after post-generation relaxation, the performance of the pretrained OMatG model and the reinforced OMatG-IRL model becomes largely similar (see Table 2).

Although post-generation relaxation eliminates the performance differences between the pretrained OMatG model and the reinforced OMatG-IRL model, it is computationally expensive. In our setup, relaxing 9046 generated structures with MACE-MPA-0 with TorchSim takes approximately $15$ minutes on an A100 GPU [and more than one day on CPU with ASE (Hjorth Larsen et al., 2017)]. From this perspective, a strong generative model for crystalline materials should not rely entirely on post-generation relaxation, but should instead produce structures that are already close to relaxed configurations. Energy-based reinforcement with OMatG-IRL moves the model in this direction by directly reducing the relative energies per atom in the generated structures.

The interpretation that energy-based reinforcement with OMatG-IRL can partially amortize relaxation into the generative model is supported by the number of LBFGS relaxation steps required in TorchSim (see Fig. 10). Compared to the pretrained OMatG model, the relaxation-step distribution for OMatG-IRL is shifted toward smaller values. In particular, the peak of the distribution decreases from 45 to 25 steps, while the median decreases from 50 to 40 steps. This indicates that OMatG-IRL produces structures that require fewer subsequent optimization steps. More broadly, these results represent a meaningful step toward generative models for crystalline materials that internalize a larger fraction of the relaxation process and rely less on expensive post-generation optimization.

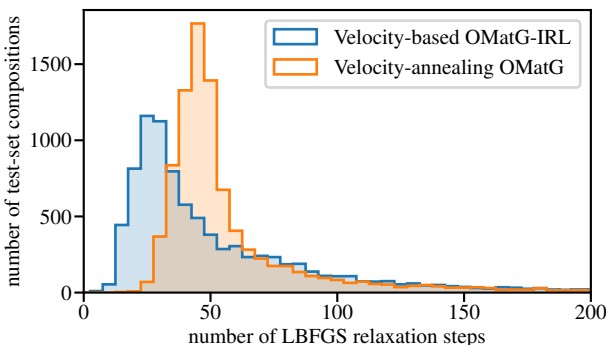

*Figure 10.* Distribution of the number of LBFGS relaxation steps required for structures generated by the pretrained velocity-annealed OMatG model and the reinforced OMatG-IRL model from Table 2 for 9046 test-set compositions. Relaxations are performed with MACE-MPA-0 in TorchSim and stopped once the maximum force falls below $0.02 \, \mathrm{eV/\mathring{A}}$, or after at most 1000 steps.

## J. Robustness of Velocity-Based OMatG-IRL

In order to assess whether the CSP improvements achieved by velocity-based OMatG-IRL in Section 5.1 are robust beyond the MP-20 dataset, we additionally evaluate the method on the MPTS-52, Alex-MP-20, and MP-20-polymorph-split datasets. The *MPTS-52* dataset (Baird et al., 2024) is a chronological split of the Materials Project containing 40 476 structures with at most 52 atoms per unit cell, while the *Alex-MP-20* dataset (Zeni et al., 2025; Höllmer et al., 2025) combines 675 204 structures with 20 or fewer atoms per unit cell from Alexandria (Schmidt et al., 2022a;b) and MP-20. We generally follow the same RL setup as in Section 5.1.2 and, importantly, use the same hyperparameters across datasets and model choices.

Table 3 summarizes the CSP performance across different datasets. Following Höllmer et al. (2025), we compare velocity-based OMatG-IRL against DiffCSP (Jiao et al., 2023), FlowMM (Miller et al., 2024), and the strongest reported OMatG models (Höllmer et al., 2025), which are established baselines for generative models of inorganic materials without explicit space-group constraints. Across all datasets, OMatG-IRL consistently improves RMSE values while remaining competitive on structure-matching metrics. At the same time, it uses far fewer integration steps and does not require a handcrafted velocity-annealing schedule.

The robustness of velocity-based OMatG-IRL across model choices is further supported in Table 4, where we apply it to different pretrained OMatG models on the MP-20 dataset, again using the same RL setup as in Section 5.1.2. These results show that velocity-based OMatG-IRL consistently improves RMSE and reduces the number of integration steps across different OMatG models, with only minor changes in structure-matching performance.

*Table 3.* Test-set evaluation metrics on the CSP task of different baselines compared to velocity-based OMatG-IRL across various datasets (best results bolded, second best underlined). The pretrained OMatG model used for each OMatG-IRL result is indicated in parentheses.

| Method | MP-20 | MPTS-52 | Alex-MP-20 | MP-20-polymorph-split |
|---|---|---|---|---|
| | match rate ↑ / RMSE ↓ / $N_t$ ↓ | match rate ↑ / RMSE ↓ / $N_t$ ↓ | match rate ↑ / RMSE ↓ / $N_t$ ↓ | METRe ↑ / cRMSE ↓ / $N_t$ ↓ |
| DiffCSP | 57.82 % / 0.0627 / 1000 | 15.79 % / 0.1533 / 1000 | - / - / - | 53.14 % / 0.279 / 1000 |
| FlowMM | 66.22 % / 0.0661 / 1000 | 22.29 % / 0.1541 / 1000 | - / - / - | 65.18 % / 0.226 / 1000 |
| OMatG | | | | |
|   Linear-ODE | **69.83** % / 0.0741 / 210 | **27.38** % / 0.1970 / 100 | 72.02 % / 0.0683 / 210 | 70.50 % / 0.187 / 950 |
|   Trig-SDE-Gamma | 68.90 % / 0.1249 / 740 | 24.51 % / 0.1867 / 740 | 72.50 % / 0.1261 / 740 | - / - / - |
|   VE-SBD-ODE | 63.79 % / 0.0809 / 660 | 21.42 % / 0.1740 / 660 | 67.79 % / 0.0674 / 660 | - / - / - |
| OMatG-IRL | 69.09 % / **0.0491** / **50** 
 (Linear-ODE) | 25.63 % / **0.1427** / **50** 
 (Linear-ODE) | **72.58** % / **0.0503** / **50** 
 (Trig-SDE-Gamma) | **70.68** % / **0.183** / **50** 
 (Linear-ODE) |

*Table 4.* MP-20 test-set CSP performance before and after applying velocity-based OMatG-IRL to different pretrained OMatG models.

| OMatG model | match rate ↑ | | | RMSE ↓ | | | $N_t$ ↓ | |
|---|---|---|---|---|---|---|---|---|
| | base | → | IRL | base | → | IRL | base → IRL | |
| Trig-SDE-Gamma | 68.90 % | → | 67.69 % | 0.1249 | → | 0.0848 | 740 → 50 | |
| Linear-ODE | 69.83 % | → | 69.09 % | 0.0741 | → | 0.0491 | 210 → 50 | |
| VE-SBD-ODE | 63.79 % | → | 63.32 % | 0.0809 | → | 0.0660 | 660 → 50 | |

# K. De Novo Generation

The main experiments in this work focus on the CSP task, where OMatG-IRL reinforces the continuous generative SI dynamics for atomic fractional positions and lattice vectors. The discrete atomic numbers are fixed and serve only as conditional inputs to the CSPNet model of OMatG. In the DNG setting, OMatG implements discrete flow matching for the generative process of the atomic numbers, typically starting from a fully masked initial configuration with unknown atomic species (Campbell et al., 2024; Höllmer et al., 2025). A complete application of policy-gradient RL to DNG would therefore require extending the policy-gradient formulation to discrete flow matching, which, to the best of our knowledge, has not yet been established. Nevertheless, OMatG-IRL can already be applied to DNG in a hybrid setting that keeps the discrete species-generation component fixed and reinforces only the continuous coordinate and lattice dynamics. We demonstrate this setup first with energy-based rewards (see Appendix K.1) and then with a symmetry-based reward (see Appendix K.2).

## K.1. Energy-Based Rewards

Analogous to the CSP experiments in Section 5.1, we first apply velocity-based OMatG-IRL to DNG with energy-based rewards. We use the best publicly available pretrained EncDec-ODE-Gamma OMatG model,[2] which was trained for the DNG task on the MP-20 dataset (Höllmer et al., 2025). As in Section 5.1, we reduce the number of integration steps from $N_t = 840$ to $N_t = 50$ for OMatG-IRL, and we remove velocity annealing from the pretrained model. Moreover, we reinforce only the generative dynamics of the atomic fractional positions. The RL hyperparameters are chosen by a sweep over the validation reward (see Appendix H). We use the square-root noise schedule with medium reference noise scale $a_{\text{ref}} = a_m$, while the exploration noise scale is now included in the hyperparameter sweep, $a_{\text{exp}} \sim \text{LogUniform}(a_s, a_l)$ (with the noise scales $a_s$, $a_m$, and $a_l$ from Fig. 2).

We consider two reward and rollout setups. In the first setup, denoted CSP-style RL, we use the pretrained OMatG DNG model in CSP mode during RL rollouts. Rather than starting from a fully masked state that is gradually unmasked during generation, the composition is fixed to one from the training dataset. All generated configurations within a GRPO group share the same composition, so we can use the reward based on the energy per atom from Section 5.1. After RL training in CSP mode, the model is switched back to DNG mode for inference, where generation again starts from a fully masked state.

In the second setup, denoted DNG-style RL, we use the DNG model directly during RL rollouts. Since different samples within a GRPO group then typically have different compositions, absolute energies are not directly comparable across the group. We therefore use the energy above hull as a composition-normalized reward. The full reward construction follows Appendix G: Invalid structures are assigned a fixed penalty value of $3\,\text{eV}$ per atom, while valid structures are assigned the negative energy above hull, such that structures closer to the convex hull receive higher rewards. Following the LeMat-GenBench protocol (Betala et al., 2025), we compute the energy above hull with MACE-MPA-0 (Batatia et al., 2025) using a self-consistent MACE-based convex hull constructed from LeMat-Bulk, a reference dataset aggregating more than $5.3$ million materials from multiple materials-science databases (Siron et al., 2025).

We evaluate DNG performance with LeMat-GenBench (Betala et al., 2025), a standardized evaluation framework for generative models of crystalline materials. We evaluate $1000$ generated structures per model (which differs from the public LeMat-GenBench leaderboard that uses $2500$ structures). The benchmark first applies a validity filter based on structural and chemical sanity checks. Stability is then assessed through the energy above hull: Structures with $E_{\text{hull}} \leq 0\,\text{eV/atom}$ are classified as stable (S), while structures with $0\,\text{eV/atom} < E_{\text{hull}} \leq 0.1\,\text{eV/atom}$ are classified as metastable (MS). Uniqueness (U) measures non-duplication within the generated set, and novelty (N) measures absence from the reference dataset, where structures are compared with Pymatgen's `StructureMatcher` (Ong et al., 2013). The SUN and MSUN rates correspond to the joint criteria $S \cap U \cap N$ and $MS \cap U \cap N$, respectively.

Table 5 compares the performance of the pretrained OMatG model to the reinforced OMatG-IRL variants. Both CSP-style and DNG-style RL achieve similar performance. They substantially improve validity, uniqueness, and novelty relative to the pretrained model. Interestingly, however, the stability-related metrics, which are directly based on the energy above hull, do not improve in the DNG setting. This contrasts with the CSP setting, where energy-based reinforcement directly improves the corresponding energy-based metrics (see Fig. 3).

We also evaluate the generated structures after post-generation relaxation with MACE-MPA-0 (Batatia et al., 2025) (see Appendix I for details on relaxation). As shown in Table 6, relaxation substantially improves the stability-related metrics for

---

[2]https://huggingface.co/OMatG/MP-20-DNG

*Table 5.* LeMat-GenBench DNG metrics for 1000 generated structures of a pretrained OMatG model ($N_t = 840$) and reinforced OMatG-IRL variants ($N_t = 50$).

| Method | Valid ↑ | Unique (U) ↑ | Novel (N) ↑ | Stable (S) ↑ | Metastable (MS) ↑ | SUN ↑ | MSUN ↑ |
|---|---|---|---|---|---|---|---|
| OMatG EncDec-ODE-Gamma | 93.4 % | 93.1 % | 52.2 % | 0.8 % | 30.0 % | 0.0 % | 3.3 % |
| CSP-style OMatG-IRL | 97.2 % | 96.9 % | 69.0 % | 0.6 % | 22.5 % | 0.0 % | 3.6 % |
| DNG-style OMatG-IRL | 97.4 % | 97.2 % | 67.5 % | 0.2 % | 21.1 % | 0.0 % | 2.0 % |

*Table 6.* LeMat-GenBench DNG metrics after relaxation for 1000 generated structures of a pretrained OMatG model ($N_t = 840$) and reinforced OMatG-IRL variants ($N_t = 50$).

| Method | Valid ↑ | Unique ↑ | Novel ↑ | Stable ↑ | Metastable ↑ | SUN ↑ | MSUN ↑ |
|---|---|---|---|---|---|---|---|
| OMatG EncDec-ODE-Gamma | 96.7 % | 96.0 % | 49.4 % | 10.6 % | 50.6 % | 0.4 % | 16.9 % |
| CSP-style OMatG-IRL | 97.9 % | 97.6 % | 64.2 % | 7.5 % | 42.2 % | 0.7 % | 18.1 % |
| DNG-style OMatG-IRL | 98.2 % | 97.7 % | 61.2 % | 7.6 % | 45.6 % | 0.6 % | 18.2 % |

all models. In this setting, both OMatG-IRL variants improve the SUN and MSUN rates relative to the pretrained model, which we view as particularly relevant metrics for inverse materials design. This improvement is primarily driven by the gains in uniqueness and novelty, which outweigh the slight decrease in stability and metastability rates.

These results show that energy-based rewards within the OMatG-IRL setup can meaningfully alter the behavior of a DNG model, even while keeping the discrete species-generation component fixed. The reinforced models also use substantially fewer integration steps $N_t$ than the pretrained OMatG model, reducing the number of steps from $N_t = 840$ to $N_t = 50$ and thus lowering inference cost by more than an order of magnitude. The fact that the final stability-oriented metrics do not improve relative to the pretrained baseline should be interpreted in light of this aggressive time discretization. For instance, at fixed low step count $N_t = 50$, DNG-style RL does improve the energy above hull from $0.89\,\mathrm{eV/atom}$ to $0.49\,\mathrm{eV/atom}$. Thus, in this DNG setting, RL substantially improves the low-step-count model but does not fully close the gap to the high-step-count pretrained baseline. This differs from the CSP setting, where OMatG-IRL not only compensates for the reduced integration budget but further improves over the pretrained model (see Fig. 3).

### K.2. Symmetry-Based Reward

We use the DNG setting to further demonstrate that velocity-based OMatG-IRL can reinforce objectives beyond energy-based rewards, including truly non-differentiable objectives. As demonstrated for velocity-annealing OMatG-IRL in Section 5.2 with the non-differentiable cRMSE-like reward, the policy-gradient RL framework treats the reward as a black box and does not require reward gradients. As an example in the velocity-based OMatG-IRL setting, which updates the full pretrained OMatG model, we encourage the generation of valid structures that are both non-triclinic and non-centrosymmetric, where validity is defined as in Appendix D and symmetry is identified with `spglib` using `symprec=0.1` (Togo et al., 2024). Non-centrosymmetric materials are technologically important because broken inversion symmetry is a prerequisite, for example, for bulk piezoelectricity and electric-dipole second-harmonic generation (Ok et al., 2006).

Starting from the reinforced DNG-style OMatG-IRL model from Appendix K.1, we add the symmetry reward alongside the energy-above-hull reward with equal weight. During RL training, the average energy above hull remains approximately constant around $0.5\,\mathrm{eV/atom}$, while the fraction of generated valid structures with non-triclinic non-centrosymmetric symmetries increases from $22.83\,\%$ to $36.79\,\%$. This provides promising evidence that OMatG-IRL can reinforce practically relevant non-differentiable objectives in addition to energy-based rewards.

## L. Details on Velocity-Annealing Reinforcement

To learn a time-dependent velocity-annealing schedule $s^\theta(t)$ (parameterized by a simple multilayer perceptron) with velocity-annealing OMatG-IRL, we first perturb the ODE dynamics of the pretrained Linear-ODE OMatG model via Eq. (14) using both constant and square-root noise schedules at different noise scales, applied to both the atomic positions and lattice vectors. Increasing the noise scale beyond a moderate range deteriorates the performance of the pretrained model in all cases. Based on these observations, we use a constant noise schedule with a reference noise scale $a_{\mathrm{ref}}$ both to define the KL-regularized reference policy and to provide exploration noise during policy rollouts (see Fig. 11).

As the reward for policy-gradient RL, we use a cRMSE-like objective (see Section 4). For each generated structure of a given composition, we identify the best-matching polymorph of the same composition in the training set. The reward is

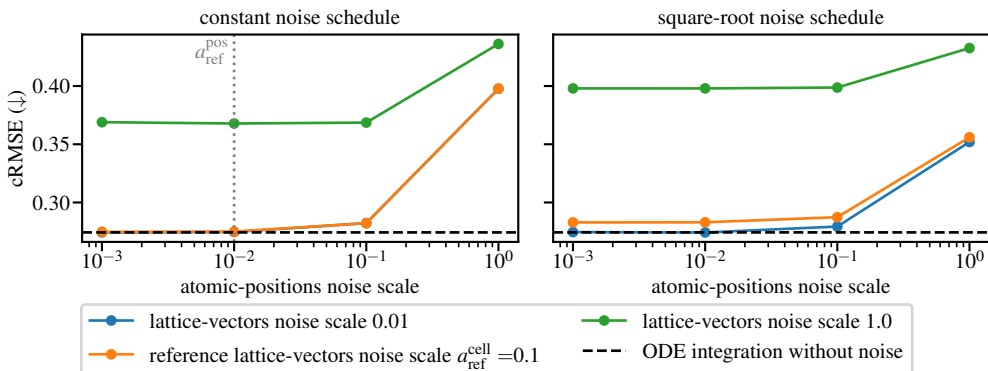

*Figure 11.* Test-set evaluation metrics for perturbed velocity-annealing ODE integration of the atomic positions and lattice vectors under different noise schedules and noise scales. The chosen reference noise scales for positions and lattice vectors are denoted by $a_{\text{ref}}^{\text{pos}}$ and $a_{\text{ref}}^{\text{cell}}$, respectively.

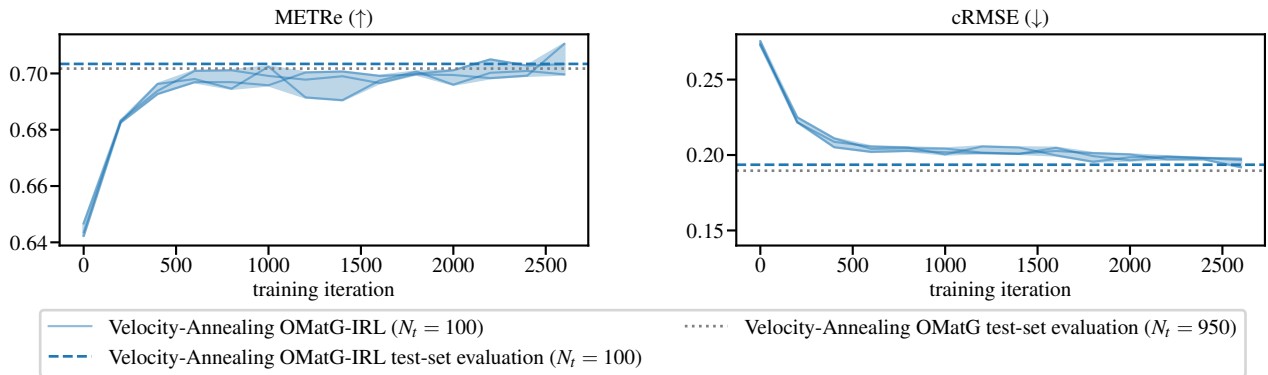

*Figure 12.* Evolution of validation metrics for velocity-annealing OMatG-IRL as a function of RL training iteration, shown for three random seeds of the same setup ($N_t = 100$). The dashed lines indicate the test-set performance of the OMatG-IRL checkpoint selected by the validation optimum. For reference, we also show the test-set performance of the original velocity-annealed OMatG model evaluated with $N_t = 950$ integration steps.

defined as $r(x_{t=1}^i) = 0.5 - \text{cRMSE}^i$, where $\text{cRMSE}^i$ is the normalized root-mean-square displacement to this best match, with missing matches penalized in the mean using `stol=0.5`. This choice ensures that lower cRMSE values correspond to higher rewards. This formulation effectively inverts the standard cRMSE computation, which identifies best matches from the generated set for each reference structure.

A hyperparameter sweep on the validation reward is used to determine the optimal configuration for velocity-annealing OMatG-IRL (see Appendix H). Notably, because this setup only rescales velocities without changing their direction, we find that explicit KL regularization is unnecessary. Velocity-annealing OMatG-IRL is able to effectively reinforce the cRMSE metric and, at its validation optimum, recovers the performance of the pretrained Linear-ODE OMatG model while using an order of magnitude fewer integration steps ($N_t = 100$ vs. $N_t = 950$; see Fig. 12).

## M. Velocity-Annealing Hyperparameter Sweep

To verify that the performance achieved by velocity-annealing OMatG-IRL at $N_t = 10$ cannot be reproduced by tuning velocity annealing alone, we perform an explicit hyperparameter sweep over the velocity-annealing parameters of the pretrained OMatG model. The sweep is conducted using Ray Tune (Liaw et al., 2018) with Optuna (Akiba et al., 2019) as the underlying sampler, and explores a uniform range between 0 and 15 for the velocity-annealing coefficients of both the atomic positions and lattice vectors.

We find that velocity-annealed OMatG at $N_t = 10$ achieves a METRe score of $65.62\%$ and a cRMSE of $0.2671$, which is substantially worse than velocity-annealing OMatG-IRL, which reaches a METRe score of $69.12\%$ and a cRMSE

of 0.2220. Despite the explicit hyperparameter sweep over handcrafted velocity-annealing schedules, no configuration recovers the performance achieved by velocity-annealing OMatG-IRL at such low integration step counts, indicating that the improvement arises from the learned annealing policy.

