# OpenReview forum: "Open Materials Generation with Inference-Time Reinforcement Learning"
_ICML.cc/2026/Conference — ICML 2026 regular_

### Official Review · Reviewer_9GhT · 2026-03-09

**Soundness:** 2
**Presentation:** 3
**Significance:** 3
**Originality:** 3
**Overall Recommendation:** 4
**Confidence:** 2

**Summary:**

This paper primarily aims to use RL to optimize the target property in crystal material generation models. Specifically, RL aligns the generation model with the downstream task/objective. The paper notes that previous methods typically require detailed scoring, while the authors propose an inference-time RL algorithm, OMatG-IRL, which directly optimizes the model without explicit score calculation. The authors applied the proposed method to crystal structure prediction tasks, achieving performance comparable to score-based methods (which may be very interesting to the community). The paper also mentions that the proposed method has high sampling efficiency and reduces generation time.

**Compliance With Llm Reviewing Policy:**

Affirmed.

**Final Justification:**

The additional results provided by the author have effectively addressed my concerns; therefore, I am very pleased to maintain my positive assessment.

**Key Questions For Authors:**

See Weakness, and:

1. Could the authors elaborate on the applicability to more flow-based generative models? This could help readers understand the applicability of the proposed method.

2. Furthermore, more ablation experiments and hyperparameter sensitivity analyses could help readers use the proposed method.

**Limitations:**

The authors could discuss this further, calculating costs in detail, broader applications (e.g., more base models), and potential social impacts.

**Strengths And Weaknesses:**

## Strengths

I found the problem this paper proposes to address is very interesting: how to achieve RL optimization when the  score is unavailable. The proposed method is concise and clear, with sufficent motivation. The writing of this paper is somewhat great.

The experimental design is reasonable, with a sufficient and relevant baseline. Furthermore, the analysis of different steps is very helpful.

Figure 1 is excellent and greatly aids in understanding the motivation and methodology of this paper.

## Weaknesses

The experimental scope of this paper could benefit from further expansion, such as trying more base models, datasets, and benchmarks. It remains unclear whether the proposed method is broadly applicable to different generative tasks.

Furthermore, this paper could benefit from broader hyperparameter sensitivity ablation and analysis, particularly the analysis of some failure scenarios, which may generate significant interest within the community.

In addition, the task and problem this paper aims to address have very broad social impact; it would be very interesting if the authors could analyze the specific potential social benefits and risks involved.

---

> ### Author Rebuttal · Authors · 2026-03-30
>
> We thank the reviewer for the favorable evaluation of the novelty, motivation, and presentation of our work, especially the observation that the score-free RL formulation may be broadly useful for the generative modeling community. Below, we address the highlighted questions and weaknesses in turn.
> # Applicability
> Our successful application of OMatG-IRL to CSP is only one contribution of this work. A central contribution is methodological rather than benchmark-specific: we introduce a velocity-based policy-gradient RL framework that avoids explicit score computation, and further show that the same idea can be used to learn velocity-annealing schedules. CSP serves as the experimental setting in which we validate the method by showing that velocity-based RL performs comparably to score-based RL.
>
> More broadly, this framework is relevant to continuous-time flow-based generative models that learn only a velocity field and do not provide an explicit score at inference time. In the materials domain, this makes the approach relevant to other flow-based crystal-generation frameworks such as FlowMM and CrystalFlow. More generally, the same idea may transfer to other continuous-time flow-based generative settings, provided that suitable task-specific rewards are available. What carries over is the core RL construction itself: introducing a stochastic surrogate process for velocity-only generative dynamics and using it to define exploration, policy-gradient updates, and reference policy for KL regularization without requiring an explicit score.
>
> We also note that hand-crafted velocity annealing was originally introduced in flow-based generative models for protein backbones, so replacing such schedules with learned inference-time policies naturally extends the relevance of our method beyond the specific OMatG setting.
>
> We will add a discussion of these applicability points (and limitations; see our reply to 82QP) to the conclusion.
> # Expanding Experimental Scope
> We agree that a broader experimental scope would strengthen the paper.
>
> In our reply to Reviewer 82QP, we show the performance of OMatG-IRL across different datasets and against additional baselines. Here, we evaluate velocity-based OMatG-IRL on three pretrained OMatG models on MP-20, in a setup analogous to Section 4.2. Values to the left of the arrow are before RL, and values to the right are after RL:
>
> |OMatG Model|Match Rate&uarr;|RMSE&darr;|$N_t$|
> |-|-|-|-|
> |Trig-SDE-Gamma|68.62% &rarr; 67.69%|0.1252 &rarr; 0.0848|740 &rarr; 50|
> |Linear-ODE|69.83% &rarr; 69.09%|0.0741 &rarr; 0.0491|210 &rarr; 50|
> |VE-SBD-ODE|63.79% &rarr; 63.32%| 0.0809 &rarr; 0.0660 | 660 &rarr; 50|
>
> These results show that velocity-based OMatG-IRL consistently improves RMSE and reduces the number of integration steps $N_t$ across different OMatG models, with only minor changes in structure-matching performance. Together with the cross-dataset comparison in our reply to Reviewer 82QP, this supports the robustness of OMatG-IRL across both model choices and datasets.
> # Ablation Experiments
> We agree that broader ablations and hyperparameter sensitivity analyses would be useful. An encouraging observation is that all results across different pretrained OMatG models and datasets were obtained with the same RL hyperparameters, suggesting that the method transfers well across both models and datasets without task-specific retuning.
>
> For the Section 4.2 setup, we further studied the role of the reference noise scale $a_\text{ref}$ and the exploration noise scale $a_\text{exp}$ in velocity-based OMatG-IRL. Holding $a_\text{exp}=a_m$ fixed, we vary $a_\text{ref}$ over the small, medium, and large noise scales $a_s$, $a_m$, $a_l$ from Fig. 2 and find that the resulting performance remains nearly unchanged. By contrast, the choice of exploration noise is much more influential. At fixed $a_\text{ref}=a_m$, the results for $a_s$, $a_m$, and $a_l$ as the exploration noise are qualitatively similar to those observed for score-based OMatG-IRL in Appendix D: $a_m$ yields the most effective reinforcement, $a_s$ leads to slower improvement due to insufficient exploration, and $a_l$ is counterproductive for energy minimization. We will add precise plots and metrics of this failure scenario in the revised manuscript.
> # Social Impact
> We agree that the social impact discussion can be made more specific and we will revise it accordingly. Our work aims to improve generative modeling for crystalline materials, which could accelerate the discovery of materials relevant to clean energy, catalysis, electronics, and other technologies. More efficient crystal generation may also reduce the cost of computational materials screening. At the same time, methods for targeted materials design may carry risks if applied irresponsibly, for example toward the design of harmful materials. In addition, large-scale model training and screening consume computational resources and therefore carry an environmental cost.

---

> > ### Author Rebuttal · Reviewer_9GhT · 2026-04-01
> >
> > The author has addressed my concerns; in light of this, I am very happy to maintain my positive assessment.

---

> > > ### Author Response · Authors · 2026-04-07
> > >
> > > We sincerely thank the reviewer for the positive assessment and thoughtful engagement with our work. We are especially encouraged that the reviewer recognized the importance of enabling policy-gradient RL even when no explicit score is available at inference time, as is the case for many flow-based generative models. We are pleased to have fully resolved all concerns and questions through the rebuttal process.
> > >
> > > We would also like to highlight that the rebuttal process across all reviews has substantially strengthened the manuscript. In particular:
> > > - We added results across multiple datasets and multiple pretrained OMatG models (see also our rebuttal to Reviewer 82QP), showing that OMatG-IRL applies robustly across different flow-based settings and can consistently improve CSP performance. These additional results further strengthen the significance of the work, and show that these improvements translate into strongly competitive CSP performance relative to relevant baselines among generative models of inorganic materials without explicit space-group constraints.
> > > - We now also demonstrate that OMatG-IRL meaningfully changes DNG behavior (see our rebuttal to Reviewer Q2j5), leading to substantial improvements in validity, uniqueness, and novelty. In addition, we added a further practically meaningful non-differentiable DNG reward (see our final rebuttal comment to Reviewer h5rs), which further supports the broader applicability of the framework.

---

### Official Review · Reviewer_Q2j5 · 2026-03-11

**Soundness:** 3
**Presentation:** 3
**Significance:** 3
**Originality:** 3
**Overall Recommendation:** 4
**Confidence:** 4

**Summary:**

Continuous-time generative models for crystalline materials can generate stable crystal structures, making them useful for inverse materials design. However, guiding these models to produce materials with specific target properties remains difficult.  To address this limitation, the paper introduces OMatG-IRL, a reinforcement learning framework designed to guide continuous-time generative models for crystalline materials toward desired properties. Traditional policy-gradient RL methods require access to the score function, which limits their use with flow-based models that only learn velocity fields. OMatG-IRL overcomes this limitation by applying policy-gradient optimization directly to the learned velocity fields during inference. The method introduces stochastic perturbations into the generation dynamics, allowing exploration while preserving the baseline performance of the pretrained model. This enables property-driven crystal structure generation without modifying the original training procedure.

**Compliance With Llm Reviewing Policy:**

Affirmed.

**Final Justification:**

I will keep my initial score of 4, and I think the reasons to accept outweigh the reasons to reject, though the concerns remain significant.

**Key Questions For Authors:**

1. The evaluation focuses only on the CSP task. Can the authors demonstrate the effectiveness of the proposed framework on de novo crystal generation as well?

2. The SI framework interpolates and integrates the continuous variables X and L in parallel through the SI dynamics. How can this framework be extended to handle the discrete variable A? Would the proposed approach still be applicable in such cases?

3. The experiments are conducted only on the MP-20 dataset. Since MPTS-52 is also a widely used benchmark for evaluating scalability in CSP tasks, could the authors provide results on this dataset to further validate the robustness and scalability of the model?

**Limitations:**

Yes

**Strengths And Weaknesses:**

***Strength***

 The paper is well written with clear motivation and sufficient background information.

 The work proposes a novel reinforcement learning framework that enables property-guided generation for flow-based crystalline material models without requiring access to the score function.

 Experimental results on benchmark datasets demonstrate the effectiveness of the approach in guiding crystal generation toward desired properties.

***Weakness***

 The evaluation is limited to the CSP task and does not include experiments on the de novo generation task.

 It is unclear how the proposed SI framework would handle the discrete variable A for de novo generation task.. While the continuous variables X and L are interpolated and integrated in parallel through the SI dynamics, extending this process to the discrete component A appears challenging.

 The experiments only use the MP-20 dataset. For CSP tasks, the MPTS-52 dataset is also widely used to evaluate model scalability, but it is not included in the experiments.

Some relevant prior works[1][2][3][4] are not cited.

[1] Wu, Hanlin, et al. "A periodic bayesian flow for material generation." arXiv preprint arXiv:2502.02016 (2025).

[2] Das, Kishalay, et al. "Periodic materials generation using text-guided joint diffusion model." arXiv preprint arXiv:2503.00522 (2025).

[3] Ding, Qianggang, Santiago Miret, and Bang Liu. "Matexpert: Decomposing materials discovery by mimicking human experts." arXiv preprint arXiv:2410.21317 (2024).

[4] Khastagir, Subhojyoti, et al. "LLM Meets Diffusion: A Hybrid Framework for Crystal Material Generation." arXiv preprint arXiv:2510.23040 (2025).

---

> ### Author Rebuttal · Authors · 2026-03-30
>
> We thank the reviewer for the positive feedback on the motivation, clarity, and novelty of our work. Below, we address the questions and concerns.
> # Discrete Flow Matching
> We agree that reinforcing species generation within OMatG would require reinforcing the discrete flow-matching component where, to the best of our knowledge, policy-gradient RL has not yet been established. It is thus beyond the scope of the present paper and remains an important direction for future work.
> # DNG
> We agree that demonstrating the effectiveness of OMatG-IRL on DNG is important. Although the current framework does not reinforce the discrete species generation, it can be applied to DNG in a hybrid setting by keeping species generation fixed and reinforcing only the continuous generative dynamics.
>
> We consider two RL setups:
> 1. In CSP-style RL, we start from the OMatG DNG model weights but use it in CSP mode within each rollout. Instead of starting from a fully masked state that is slowly unmasked by discrete flow matching, the model keeps the composition fixed to one from the training dataset. This lets us apply the Section 4.2 procedure directly and use the same energy-based reward, since all samples in a GRPO group share the same composition. After RL in CSP mode is finished, the model is switched back to standard DNG mode, i.e., inference starts at the fully masked state again.
> 2. In DNG-style RL, we use the DNG model directly during RL. Since different samples in a GRPO group generally have different compositions, absolute energy is not comparable across the group. Instead, we use energy above hull as a composition-normalized reward.
>
> To evaluate DNG performance, we use LeMat-GenBench which provides a standardized evaluation framework and broad leaderboard. The following table compares our best pretrained OMatG DNG model and the reinforced variants to the current leaderboard leaders, only considering direct model outputs without post-relaxation:
>
> |Method|Valid&uarr;|Unique (U)&uarr;|Novel (N)&uarr;|Stable (S)&uarr;|Metastable (MS)&uarr;|SUN&uarr;|MSUN&uarr;|
> |-|-|-|-|-|-|-|-|
> |Leader|96.8% (OQMD)|96.4% (OQMD)|**69.4%** (CrystalFlow)|**9.3%** (AFLOW)|**36.5%** (ADiT)|**0.3%** (CrystaLLM-pi)|**8.5%** (DiffCSP)|
> |OMatG EncDec-ODE-Gamma|93.4%|93.1%|52.2%|0.8%|30.0%|0.0%|3.3%|
> |OMatG-IRL (CSP)|**97.2%**|**96.9%**|69.0%|0.6%|22.5%|0.0%|3.6%|
> |OMatG-IRL (DNG)|**97.4%**|**97.2%**|67.5%|0.2%|21.1%|0.0%|2.0%|
>
> Both reinforced variants substantially improve validity, which likely also boosts uniqueness and novelty, since validity acts as a pre-filter in LeMat-GenBench. This agrees with our results in the CSP setting where we saw a significant reduction of invalid structures. However, the stability-related metrics do not directly improve here.
>
> The following table shows the corresponding results after MACE relaxation, now including leaderboard models that report relaxed-structure evaluations:
>
> |Method|Valid&uarr;|Unique (U)&uarr;|Novel (N)&uarr;|Stable (S)&uarr;|Metastable (MS)&uarr;|SUN&uarr;|MSUN&uarr;|
> |-|-|-|-|-|-|-|-|
> |Leader|96.8% (OQMD)|96.4%(OQMD)|**70.5%** (MatterGen)|**12.4%** (PLaID++)|**60.7%** (PLaID++)|**1.0%** (PLaID++/OMatG)|18.0% (OMatG)|
> |OMatG EncDec-ODE-Gamma|96.7%| 96.0%|49.4%|10.6%|50.6%|0.4%|16.9%|
> |OMatG-IRL (CSP)|**97.9%**|**97.6%**|64.2%|7.5%|42.2%|0.7%|**18.1%**|
> |OMatG-IRL (DNG)|**98.2%**|**97.7%**|61.2%|7.6%|45.6%|0.6%|**18.2%**|
>
> Both OMatG-IRL variants noticeably improve MSUN compared to the pretrained model, which we view as one of the most informative aggregate metrics.
>
> These results are preliminary and not fully matched to the leaderboard protocol yet since we evaluate 1000 rather than 2500 generated structures. Also, in contrast to the OMatG leaderboard entry, we use MACE rather than UMA relaxation. Still, these results are promising. Our hybrid setup does not require explicit diversity rewards because the species-generation component is fixed during RL, and it achieves leaderboard-level performance.
>
> We will discuss and include the final results in the revised manuscript.
> # Additional Datasets
> We agree that evaluation beyond MP-20 would strengthen the paper. In our replies to Reviewers 82QP and 9GhT, we show results for OMatG-IRL on the MPTS-52 and Alex-MP-20 datasets, as well as across different pretrained OMatG models on MP-20. Together, the results support the robustness and scalability of OMatG-IRL.
> # Prior Work
> We thank the reviewer for pointing out these references. Our current related-work discussion in Section 1.1 focuses on continuous-time generative models for crystalline materials that operate directly on atomic positions, lattice parameters, and atom types without explicit space-group constraints, while referring readers to De Breuck et al. for broader coverage. In the revised manuscript, we will broaden this discussion to include works such as TGDMat, CrysBFN, and CrysLLMGen in this context, and add a short discussion of more LLM-centric materials-discovery frameworks such as MatExpert.

---

> > ### Author Rebuttal · Reviewer_Q2j5 · 2026-04-03
> >
> > Thank you to the authors for the thoughtful rebuttal. However, the model’s performance on the DNG task is quite poor, particularly on the stability and metastability metrics. Therefore, I will retain my score.

---

> > > ### Author Response · Authors · 2026-04-07
> > >
> > > We thank the reviewer for the follow-up remark and for acknowledging that our rebuttal addressed the main concerns raised in the initial review.
> > >
> > > # Interpretability of DNG Results
> > >
> > > We would like to emphasize that the DNG experiment added during the rebuttal period should not be overinterpreted. We do not present it as evidence that OMatG-IRL outperforms all previous DNG methods across all metrics. Rather, we include it as a test of whether the OMatG-IRL framework can extend beyond the main CSP setting of the manuscript and meaningfully change generative behavior in this setting.
> > >
> > > In that respect, we believe the additional results are informative and provide evidence that the framework is not narrowly tied to CSP. Comparing the pretrained OMatG EncDec-ODE-Gamma model with the corresponding OMatG-IRL model reinforced with a DNG-style energy-above-hull reward in the DNG results table for relaxed structures from our initial rebuttal shows the following aggregate changes in key DNG metrics during RL training:
> > >
> > > |Valid&uarr;|Unique (U)&uarr;|Novel (N)&uarr;|Stable (S)&uarr;|Metastable (MS)&uarr;|SUN&uarr;|MSUN&uarr;|
> > > |-|-|-|-|-|-|-|
> > > |+1.5%|+1.7%|+11.8%|-3.0%|-5.0%|+0.2%|+1.3%|
> > >
> > > While the stability-oriented metrics decrease, the validity, uniqueness, novelty, SUN, and MSUN metrics all improve. We therefore view this as evidence that even a straightforward energy-above-hull reward can meaningfully change the behavior of the pretrained OMatG model in the DNG setting. In particular, the improvement in SUN and MSUN is encouraging, since these metrics capture the joint trade-off between novelty, uniqueness, and stability that is central in materials design.
> > >
> > > We would also like to highlight that the reinforced model uses only $N_t = 50$ integration steps, compared to $N_t = 840$ for the pretrained model, corresponding to more than an order-of-magnitude faster generation. Importantly, when considering the fixed integration-step count $N_t = 50,$ RL does improve the underlying reward. The energy above hull per atom decreases from 0.89 eV at the start of RL to 0.49 eV at the end. We therefore view this as further evidence that OMatG-IRL meaningfully reinforces the target objective.
> > >
> > > The fact that the final stability-oriented metrics do not improve relative to the pretrained model evaluated at $N_t = 840$ is due to the deterioration of the pretrained model under such aggressive time discretization, rather than a failure of RL to improve the reward at fixed $N_t$. In this DNG setting, RL improves the low-step-count model substantially, but does not fully close the gap to the much more expensive pretrained baseline. Determining how best to overcome this remaining gap is beyond the scope of this rebuttal period.
> > >
> > > # DNG with Non-Differentiable Reward
> > >
> > > Encouraged by the reviewer’s concern about broader utility in DNG and a suggestion by Reviewer h5rs, we also explored a practically meaningful objective in the DNG setting. Specifically, we considered a non-differentiable symmetry-based reward that encourages the generation of structures that are both non-triclinic and non-centrosymmetric, as identified by `spglib` up to tolerance. Non-centrosymmetric materials are technologically important because broken inversion symmetry is a prerequisite, for example, for bulk piezoelectricity and electric-dipole second-harmonic generation.
> > >
> > > Starting from the OMatG-IRL DNG model obtained in our previous rebuttal, we add this symmetry reward on top of the energy-above-hull-based objective. During RL training, the average energy above hull per atom remains roughly constant around 0.5 eV, while the fraction of generated structures with non-trivial non-centrosymmetric symmetries increases from 22.83% to 36.79%. We view this as promising preliminary evidence that OMatG-IRL can be used not only to shift DNG behavior in terms of validity, uniqueness, and novelty, but also to reinforce practically meaningful objectives in a targeted way.
> > >
> > > # Summary
> > >
> > > Taken together, we believe the rebuttal process across all reviews has substantially strengthened the manuscript. In particular:
> > >
> > > - We broadened the CSP evidence by adding results across multiple datasets, multiple pretrained OMatG models, and additional baseline comparisons. These new experiments show that OMatG-IRL remains robust across different flow-based settings, consistently improves CSP performance, and yields strongly competitive results relative to relevant baselines (see our rebuttals to Reviewers 82QP and 9GhT).
> > > -  We extended the rebuttal beyond CSP by showing that OMatG-IRL can also induce meaningful changes in DNG behavior under an energy-based objective, even though DNG is not the main focus of the manuscript.
> > > -  We further strengthened the DNG experiment by adding a symmetry-driven, explicitly non-differentiable DNG reward, showing that the framework can accommodate practically meaningful black-box objectives.

---

### Official Review · Reviewer_h5rs · 2026-03-13

**Soundness:** 3
**Presentation:** 3
**Significance:** 3
**Originality:** 3
**Overall Recommendation:** 5
**Confidence:** 4

**Summary:**

This work introduces how policy-gradient reinforcement learning can be applied towards models that fall under the general stochastic interpolant framework, including models that correspond to ODEs without an explicit score function, and models that correspond to SDEs. In the ODE case, this is accomplished by adding small noise perturbations to the ODE of the base model, providing an SDE that can be used as a reference policy for GRPO. Additionally, this work introduces a learned inference-time velocity-annealing schedule to improve performance.
This framework is applied to the task of crystal structure prediction, using pretrained stochastic interpolant models from OMatG and applying reinforcement learning to minimize the energies of predicted structures.

**Compliance With Llm Reviewing Policy:**

Affirmed.

**Final Justification:**

The paper is well presented, solving an interesting problem using a method that is of interest to the broader community, not just for materials generation. The authors show strong results in the application of this method towards the task of CSP – however the impact of this improvement may be overstated and the interpretation of these results can be better explained, which the authors have promised to do.
After responding to reviewers, the authors have extended the work significantly to include more experiments. This has improved the manuscript greatly, showing that the proposed method has many potential uses.

**Key Questions For Authors:**

Questions:
1. How does the CSP performance of this method compare to just applying relaxation to the structures of the pretrained model, using the same energy-prediction model?
2. How does CSP performance compare to applying energy-based guidance?
3. For what other rewards could this method be used for?

Suggestion:

  4. It would be helpful to include a table that summarized the results, showing the performance of the baseline SI methods along with the change given by different noising scales and schedules. Ideally this would include multiple baseline SI methods with different interpolants, to show how useful the method is in practice.

  5. It would be helpful to present the computational cost of this method.

  6. It would be helpful to include an explicit comparison to the method of Liu et al. (2023), e.g. by showing their equation for the linear interpolant case.

**Limitations:**

- The added computational cost of this method is not discussed.
- While CSP is an important task, this method is very limited by only being applicable to CSP and not to DNG, a task that would include a larger set of relevant rewards.

**Strengths And Weaknesses:**

### Strengths
- Inference-time reinforcement learning can be very useful for generative models, and this work opens its application up to a broader class of generative models by demonstrating how it can be applied even when there isn't any access to a score function. This is a novel and useful contribution. This work does a good job at comparing the performance of RL using a score-based SDE and RL using a perturbed ODE at different noise scales.
- The application of energy-based reinforcement learning to optimize crystal structure prediction is novel to my knowledge. Strong results are shown in terms of reducing the number of integration steps necessary to perform accurate CSP.
- The background on Stochastic interpolants as well as policy-gradient RL is well presented.

### Weaknesses
- The presentation of the results are not clear. The paper never directly presents all metrics for a baseline model along with the metrics for the model after reinforcement learning on energy. This makes it harder to evaluate results.
-  The application of this method to crystal structure prediction seems to be of limited practical use, as it is currently presented. Energy is something for which we do have access to gradients, so these could be incorporated as guidance. While the RMSE and relative energy per atom seem to be improved by this method, the match rate doesn't seem to improve. It is possible that this method is using the energy reward as a way to reach an exact local energy minima, but not predicting different polymorphs. In practice, the crystal structures predicted by CSP models are usually further relaxed with MLIPs or with DFT, so the results after relaxation might be the same.

---

> ### Author Rebuttal · Authors · 2026-03-30
>
> We thank the reviewer for the encouraging feedback on the novelty and usefulness of our framework. Below, we address the questions and suggestions in turn.
> # Relaxation
> We agree that comparison to post-generation relaxation is important, and we will include such results in the revised manuscript. In short, for the Section 4.2 setup, the performance of OMatG and OMatG-IRL becomes much more similar after relaxation. This is illustrated for one OMatG-IRL model below. Values to the left of the arrow are before relaxation, and values to the right are after relaxation:
> |Method|Match Rate&uarr;|RMSE&darr;|relative energy per atom / eV&darr;|invalid-energy rate&darr;|
> |-|-|-|-|-|
> |Velocity-Annealing OMatG test-set evaluation ($N_t=740$)|68.62% &rarr; 68.27%|0.1252 &rarr; 0.0550|0.6529 &rarr; 0.0290|12 / 9046 &rarr; 0 / 9046|
> |Velocity-Based OMatG-IRL with ref. noise scale $a_s$ ($N_t=50$)|67.88% &rarr; 67.36%|0.0825 &rarr; 0.0604|0.1797 &rarr; 0.0240|14 / 9046 &rarr; 0 / 9046|
>
> However, the computational cost of relaxation is substantial. Relaxing 9,046 structures with MACE takes about 15 hours on CPU with ASE, and 15 minutes on an A100 GPU with TorchSim. From our perspective, a strong generative model should directly produce well-relaxed structures, and OMatG-IRL pushes CSP models in this direction while also significantly reducing the number of integration steps.
> # Energy-Based Guidance
> We agree that inference-time energy-gradient guidance is an interesting alternative, especially when a differentiable surrogate energy model is available. However, one motivation for OMatG-IRL is that it does not require reward gradients and therefore also applies to black-box rewards, as illustrated by the cRMSE-like reward in Section 4.3.
>
> The role of energy-based RL is also broader than simply steering individual trajectories at inference time. Our approach reinforces the OMatG backbone itself so that it more consistently generates lower-energy configurations. For a general objective such as energy minimization, we view this as an improvement of the generative model itself. In this sense, the usual drawback that RL must be repeated for each downstream reward is less central in the present setting.
>
> A comparison to energy-based guidance would still be very interesting. However, to the best of our knowledge, such approaches have not yet been systematically explored for generative models for inorganic crystals. A careful comparison would thus require substantial additional development and evaluation, and is beyond the scope of the present manuscript.
>
> We will add a discussion of these points to the conclusion.
> # Rewards
> CSP with an energy-based reward is only one instance of the broader OMatG-IRL framework. In fact, the velocity-annealing experiments in Section 4.3 already use a cRMSE-like reward, showing that OMatG-IRL is not restricted to a single objective.
>
> More generally, the framework can be used with any black-box reward defined on the generated structure. Within CSP, one could imagine rewards related to symmetry or space-group consistency, which could move OMatG closer to generative models with explicit space-group constraints. Within DNG, one could in principle also consider application-specific rewards such as band gap or bulk modulus.
>
> For a broader discussion of the applicability of our method to other flow-based generative models, we refer to our reply to Reviewer 9GhT.
> # Robustness
> We agree that an explicit summary table for the results in Figs 3 and 4 would be helpful. We also agree that including more OMatG models would underline the robustness of OMatG-IRL, and we will add this to the revised manuscript.
>
> In our reply to Reviewer 9GhT, we show results for different pretrained OMatG models on the MP-20 dataset. In our reply to Reviewer 82QP, we show performance of OMatG-IRL across different datasets including other baselines. In summary, OMatG-IRL robustly improves performance across different OMatG models and datasets.
> # Computational Cost
> We agree that the computational cost should be discussed more explicitly. There are two aspects. First, RL introduces an additional training-stage cost. For example, with batched energy evaluation using TorchSim on an A100 GPU, training the models in Fig. 3 takes roughly 6 hours, compared to more than a day for training the original OMatG models.
>
> More importantly, evaluating the trained OMatG-IRL model remains essentially as fast as evaluating the pretrained OMatG model. In the velocity-annealing case, the only extra component is a lightweight MLP that is negligible compared to OMatG's GNN. Also, OMatG-IRL consistently reduces the number of required integration steps, which leads to substantially faster generation overall.
> # Comparison to FlowGRPO
> We agree that a more explicit discussion of FlowGRPO's approach would be helpful and we will include this in the revised manuscript.
> # DNG
> In our reply to Reviewer Q2j5, we show results for the application of OMatG-IRL to DNG.

---

> > ### Author Rebuttal · Reviewer_h5rs · 2026-04-01
> >
> > I thank the authors for responding to the points I have raised. The new experiments do a good job of showing the limitations and generalizability of the method and puts its performance in the context of other methods.
> >
> >
> > A few more suggestions:
> > - The results from the post-relaxation experiment confirms that the model is essentially pre-relaxing crystals rather than discovering new polymorphs. I agree that this is still useful, since it is amortizing the cost of evaluating an MLIP. I think this aspect should be emphasized in the paper. One way to demonstrate that this is useful is to show whether the structures generated by OMatG-IRL need fewer relaxation steps to converge compared to the structures generated by OMatG.
> > - A DNG experiment with a non-differentiable reward (such as bandgap) would help demonstrate the utility of the method and could increase the impact of this paper.
> > - Subsection 2.4 might be better suited as a subsection of section 4.

---

> > > ### Author Response · Authors · 2026-04-07
> > >
> > > We sincerely thank the reviewer for the continued engagement with our work and for indicating that the main concerns have been fully addressed through the rebuttal process. We are especially encouraged that the additional experiments were found to clarify both the limitations and the broader applicability of the method, while also putting its performance in the context of other approaches. Below, we address the suggestions in turn.
> > >
> > > # Fewer Relaxation Steps
> > >
> > > We agree that explicitly showing a reduction in relaxation steps would be valuable, and we are happy to include this analysis in the revised manuscript. For the pretrained OMatG model and the reinforced OMatG-IRL model from the table in our initial rebuttal, we find that the distribution of LBFGS relaxation steps in TorchSim is noticeably shifted toward smaller values for OMatG-IRL. In particular, the peak of the distribution shifts from 45 to 25 steps, corresponding to a reduction of 44%, while the mean decreases from 50 to 40 steps, corresponding to a reduction of 20%. Both distributions retain a similar long tail up to the maximum of 100 relaxation steps.
> > >
> > > We agree with the reviewer that this supports the interpretation that OMatG-IRL can effectively amortize part of the relaxation process, which is practically useful in its own right. More broadly, however, our ultimate goal is to move as much of this expensive relaxation as possible into the generative model itself, and we view these results as a meaningful step in that direction.
> > >
> > > # DNG with Non-Differentiable Reward
> > >
> > > We thank the reviewer for this suggestion and agree that such an experiment is valuable for further demonstrating the utility of the method. Importantly, our RL framework treats the reward as a black box and does not require access to gradients, so it is in principle compatible with arbitrary non-differentiable objectives. This is one of the main motivations for using policy-gradient RL in the first place.
> > >
> > > We performed a DNG experiment with a non-differentiable reward that is meaningful in our OMatG-IRL setting. Specifically, we encourage the generation of structures that are both non-triclinic and non-centrosymmetric, as identified by `spglib` up to tolerance. Non-centrosymmetric materials are technologically important because broken inversion symmetry is a prerequisite, for example, for bulk piezoelectricity and electric-dipole second-harmonic generation.
> > >
> > > In our experiment, we start from the OMatG-IRL DNG model obtained in our rebuttal to Reviewer Q2j5 and add an additional symmetry reward that encourages the generation of non-triclinic, non-centrosymmetric structures. During RL training, the average energy above hull per atom remains roughly constant around 0.5 eV, while the fraction of generated structures with non-trivial non-centrosymmetric symmetries increases from 22.83% to 36.79%.
> > >
> > > We note that we did not choose bandgap for this experiment because, in our current OMatG-IRL setup, bandgap targeting would likely require stronger control over species generation and an additional surrogate-property model. We nevertheless view our new results as promising evidence that OMatG-IRL can reinforce practically relevant non-differentiable objectives. We would be happy to include these results in the revised manuscript.
> > >
> > > # Move Subsection 2.4
> > >
> > > We thank the reviewer for this suggestion and agree that this may improve the presentation. We will consider moving Subsection 2.4 to Section 4 in the revised manuscript.
> > >
> > > # Summary
> > >
> > > We believe that the rebuttal process across all reviews has substantially strengthened the manuscript. In particular:
> > >
> > > - We addressed the request for broader experiments by adding results across multiple datasets, multiple pretrained OMatG models, and additional baseline comparisons. These results show that OMatG-IRL applies robustly across different flow-based settings, consistently improves CSP performance, and yields strongly competitive CSP results relative to relevant baselines (see our rebuttals to Reviewers 82QP and 9GhT).
> > > - We explicitly showed that OMatG-IRL reduces the number of relaxation steps in the CSP task.
> > > - We showed that OMatG-IRL can meaningfully reinforce DNG behavior (see also our rebuttal to Reviewer Q2j5) and added a further relevant DNG reward to demonstrate compatibility with practically meaningful non-differentiable objectives.

---

### Official Review · Reviewer_82QP · 2026-03-13

**Soundness:** 3
**Presentation:** 2
**Significance:** 3
**Originality:** 3
**Overall Recommendation:** 3
**Confidence:** 2

**Summary:**

The authors propose using inference-time reinforcement learning for inverse materials design, called OMatG-IRL. They introduce a policy-gradient RL framework to avoid score computation and benchmark their model on a CSP task (MP-20 dataset).

**Compliance With Llm Reviewing Policy:**

Affirmed.

**Key Questions For Authors:**

1) Do you have any experiments that help readers compare the performance of your model to previous models?
2) You are using simple multilayer perceptrons; have you considered more advanced models like GNNs or transformers?
3) Even if your model turns out to be less performant compared to previous work, your proposed framework might still be relevant and a strong contribution. Can you state the limitations of your work more clearly?

**Limitations:**

The limitations are not clearly stated, and this is the main drawback in my opinion. Does this work provide an interesting framework that underperforms because of architectural limitations, or is it competitive with previous works?

**Strengths And Weaknesses:**

Soundness: The motivation and proposed RL framework seem relevant. The dataset and metrics are appropriate for this task. However, it should be noted that the authors use a single, small dataset (MP-20) for their experiments and do not compare their model to baselines, even though several baselines for CSP tasks exist.
Presentation: The presentation is detailed and the figures are relevant and helpful in understanding the submission. However, the RL framework could be introduced more gently at the beginning to ease reading of section 3; other sections are better introduced.
Significance: This research topic is important for material generation because inverse design remains a challenging task. But the experiments proposed by the authors make comparison with previous work difficult, and it is not clear to me how their work performs relative to prior work, even if the contributions might be very relevant.
Originality: The approach is novel for the CSP task and is a valuable contribution to the development of RL in materials science.

---

> ### Author Rebuttal · Authors · 2026-03-29
>
> We thank the reviewer for the positive assessment of the soundness and originality of our work, and for the constructive feedback on how to strengthen its significance and presentation. Below, we address the three questions.
> # Baselines and Datasets
> We agree that the significance can be strengthened by clearer comparison to prior CSP models and by showing that OMatG-IRL is effective beyond a single dataset. In the revised manuscript, in addition to MP-20 and MP-20-Polymorph-Split (45,229 structures each), we will consider the standard datasets MPTS-52 (40,476 structures) and Alex-MP-20 (675,204 structures). As baselines, following the OMatG paper, we will consider DiffCSP, FlowMM, and the best OMatG models. These are established baselines for generative models of inorganic materials without explicit space-group constraints. In brief, the results are as follows (best results bolded, second-best italicized, and the metrics are introduced in Section 2.4):
>
> |Method|MP-20 (Match Rate&uarr; / RMSE&darr; / $N_t$&darr;)|MPTS-52 (Match Rate&uarr; / RMSE&darr; / $N_t$&darr;)|Alex-MP-20 (Match Rate&uarr; / RMSE&darr; / $N_t$&darr;)|MP-20-Polymorph-Split (METRe&uarr; / RMSE&darr; / cRMSE&darr; / $N_t$ &darr;)|
> |-|-|-|-|-|
> |DiffCSP|57.82% / *0.0627* / 1000|15.79% / *0.1533* / 1000|- / - / -|53.14% / 0.084 / 0.279 / 1000|
> |FlowMM|66.22% / 0.0661 / 1000|22.29% / 0.1541 / 1000|- / - / -|65.18 % / 0.079 / 0.226 / 1000|
> |OMatG Linear-ODE|**69.83%** / 0.0741 / *210*|**27.38%** / 0.1970 / *100*|72.02% / 0.0683 / *210*|*70.50%* / *0.056* / *0.187* / *950*|
> |OMatG Trig-SDE-Gamma|68.90% / 0.1249 / 740|24.51% / 0.1867 / 740|*72.50%* / 0.1261 / 740|- / - / - / -|
> |OMatG VE-SBD-ODE|63.79% / 0.0809 / 660|21.42% / 0.1740 / 660|67.79% / *0.0674* / 660|- / - / - / -|
> |OMatG-IRL|*69.09%* / **0.0491** / **50** (Linear-ODE)|*25.63%* / **0.1427** / **50** (Linear-ODE)|**72.58%** / **0.0503** / **50** (Trig-SDE-Gamma)|**70.68%** / **0.051** / **0.183** / **50** (Linear-ODE)|
>
> OMatG-IRL consistently improves RMSE across all datasets while remaining competitive on structure-matching metrics. It also requires far fewer integration steps $N_t$ and no hand-crafted velocity-annealing schedule.
>
> # Model Architecture
> We will revise Sections 3.1 and 3.2 to make clear that, in both the score-based and velocity-based settings, OMatG-IRL updates the full pretrained OMatG backbone. This backbone is not a simple MLP. As described in Section 2.1, OMatG uses CSPNet, an E(n)-equivariant GNN.
>
> We only use a simple MLP in the velocity-annealing OMatG-IRL variant in Section 3.3. There, the pretrained OMatG backbone is frozen, and RL learns only a time-dependent residual schedule that rescales the frozen velocity field. Because this schedule depends only on the one-dimensional time variable, rather than on the full crystal configuration, we chose a simple MLP as the appropriate architecture.
>
> More generally, we will expand the introduction to Section 3 to clarify the relation between the three variants. In particular, we will explain more explicitly that score-based OMatG-IRL is the direct policy-gradient RL formulation when both the velocity field and denoiser/score are available, velocity-based OMatG-IRL is our new extension that enables policy-gradient RL when only the velocity field is available at inference time, and velocity-annealing OMatG-IRL is a closely related variant in which RL learns a time-dependent annealing schedule for the velocity field.
> # Limitations
> We agree that the conclusion would benefit from a more explicit discussion of limitations (and applicability; see our reply to Reviewer 9GhT). In summary:
> - The current framework is designed for continuous generative dynamics and is not directly applicable to the discrete flow-matching component of OMatG. However, the de novo generation (DNG) task can still be considered by keeping the discrete species-generation component frozen while reinforcing the continuous generative dynamics of the positions and/or lattice vectors (see our reply to Reviewer Q2j5).
> - Velocity-based OMatG-IRL relies on the surrogate stochastic process in Eq. (13), whose marginals deviate (in a controlled manner) from those of the score-based stochastic process in Eq. (6). As shown in Fig. 2, the surrogate process supports only smaller noise scales than the score-based process. In our experiments, the noise scale that yields the best RL performance remains accessible in the velocity-based setting (see Fig. 6), but this may not always be the case. Empirically, this limitation appears moderate. In our case, smaller accessible exploration noise mainly slows convergence, whereas clearly worse performance is observed only at overly large noise scales (see Figs 3 and 5).
> - Our energy-based reward substantially improves the relative energy per atom and RMSE in the CSP task, but does not directly optimize structure-matching metrics such as match rate or METRe. Developing rewards that improve these metrics remains an open question.

---

### Decision · Program_Chairs · 2026-04-30

**Decision:**

Accept (regular)

**Comment:**

The paper tackles an important and timely problem, i.e., enabling inference-time reinforcement learning for inverse materials design without requiring access to score functions. Reviewers generally agree that the proposed framework is novel and technically interesting, with clear motivation, a solid methodological formulation, and promising improvements demonstrated on the CSP task. The rebuttal successfully addressed major concerns raised by Reviewer h5rs (accept), Reviewer Q2j5 (weak accept), and Reviewer 9GhT (weak accept), who eventually reached a consensus toward accepting the paper.

Reviewer 82QP (weak reject) raised concerns that the experimental evaluation was limited to a single small dataset (MP-20) and lacked comparisons with existing CSP baselines, making it difficult to assess performance relative to prior work. Additionally, while the presentation is generally clear, the RL framework could be introduced more gradually to improve readability. The reviewer also noted the lack of a thorough discussion of limitations. Overall, despite the novelty and importance of the approach, the initial submission did not provide sufficient evidence to convincingly demonstrate its effectiveness compared to existing methods. In the rebuttal, the authors provided additional experimental results with more baselines and datasets, clarified the model architecture, and included a more explicit discussion of limitations. These revisions directly addressed the main concerns raised. Although Reviewer 82QP did not provide a post-rebuttal update, the AC confirms that the rebuttal has sufficiently resolved the key issues.

Overall, the AC believes that all major concerns have been adequately addressed and agrees that the paper presents a promising and novel framework with meaningful potential impact. The authors are encouraged to incorporate the additional results and clarifications from the rebuttal into the final version to further strengthen the paper.